# Non-apoptotic caspase-8 is critical for orchestrating exaggerated inflammation during severe SARS-CoV-2 infection

Inflammation and excess cytokine release are hallmarks of severe COVID-19. While programmed cell death is known to drive inflammation, its role in SARS-CoV-2 pathogenesis remains unclear. Using gene-targeted murine COVID-19 models, we here find that caspase-8 is critical for cytokine release and inflammation. Loss of caspase-8 reduces disease severity and viral load in mice, and this occurs independently of its apoptotic function. Instead, reduction in SARS-CoV-2 pathology is linked to decreased IL-1β levels and inflammation. Loss of pyroptosis and necroptosis mediators in gene-targeted animals provides no additional benefits in mitigating disease outcomes beyond that conferred by loss of caspase-8. Spatial transcriptomic and proteomic analyses of caspase-8-deficient mice confirm that improved outcomes are due to reduced pro-inflammatory responses, rather than changes in cell death signalling. Elevated expression of caspase-8 and cFLIP in infected lungs, alongside caspase-8-mediated cleavage of N4BP1, a suppressor of NF-kB signalling, indicates a role of this signalling axis in pathological inflammation. Collectively, these findings highlight non-apoptotic functions of caspase-8 as a driver of severe COVID-19 through modulation of inflammation, not through the induction of apoptosis.

During COVID-19, dysregulated hyperinflammatory immune responses can lead to severe disease and mortality. Despite substantial progress in understanding SARS-CoV-2 induced pathology, specific molecular events that ultimately trigger severe inflammation and fatalities remain poorly understood. Unravelling the molecular mechanisms that contribute to the immune dysregulation following SARS-CoV-2 infection is paramount in formulating effective strategies to mitigate mortality and morbidity with longer-term repercussions associated with this virus.

SARS-CoV-2 has been reported to trigger the activation of critical components of the host cell death machinery in diverse cell types[1,2]. Pyroptosis is a distinct form of programmed cell death characterized by necrotic features and an inflammatory response (reviewed in ref. 3). It is triggered by the activation of inflammatory caspases-1 (human and mouse), −4 (human), −5 (human), or −11 (mouse)[4], which can proteolytically activate the pore-forming protein gasdermin D (GSDMD)[5].

Activated caspase-1 also cleaves the pro-inflammatory cytokines interleukin-1β (IL-1β) and interleukin-18 (IL-18) into their active forms[4,6]. Necroptosis is another lytic form of programmed cell death. It triggers the release of damage associated molecular patterns (DAMPs), provoking immune activation, cytokine release and inflammation[5,7]. Necroptosis can be activated via death receptor signalling, such as stimulation of tumour necrosis factor (TNF) receptor 1 (TNFR1)[8,9], the cytoplasmic nucleic acid sensor Z-DNA binding protein 1 (ZBP1)[10,11] and downstream of Toll-like receptors (TLR)[12,13]. In the absence of catalytically active caspase-8[14], these signals lead to the activation of Receptor-Interacting Protein Kinase 3 (RIPK3) which licenses the pseudokinase mixed lineage kinase domain-like (MLKL) to effect plasma membrane rupture[15–17]. Although necroptosis and pyroptosis have been linked to the release of DAMPs and cytokines during SARS-CoV-2 infection[18–21], our previous genetic studies showed that these processes are not required for severe COVID-19 and excess levels of cytokines in mice[22,23].

e-mail: bader.s@wehi.edu.au; m.pellegrini@centenary.org.au; doerflinger.m@wehi.edu.au

Apoptotic cell death is essential for the removal of damaged, superfluous, or infected cells. It is characterized by the coordinated degradation of intracellular content, generating contained apoptotic vesicles. Apoptosis was long thought to be immunologically silent, however, recent findings have implicated a role for some of its regulators in the modulation of inflammatory responses[24–27]. Apoptosis can be initiated through the intrinsic (mitochondrial) or extrinsic pathway (death receptor activated) pathways[3]. The intrinsic pathway is activated by stress stimuli and is controlled by pro- and anti-apoptotic members of the BCL-2 protein family[28]. Extrinsic apoptosis occurs when death ligands (e.g. TNF) bind to their cognate death receptors (e.g. TNFR1), leading to the activation of the initiator caspase, caspase-8 (in humans also caspase-10). This process is regulated by cellular inhibitor of apoptosis proteins (cIAPs), which prevent the activation of caspase-8 and can drive NF-κB signalling. Through the induction of target genes, NF-κB promotes cell survival and production of inflammatory cytokines[29]. Both intrinsic and extrinsic apoptotic cell death pathways converge on the activation of the executioner caspases −3 and −7, which drive a cascade of proteolytic events that culminate in the demolition of the cell[30].

Recent studies have broadened our understanding of the role of caspase-8 beyond apoptosis, including functions such as cleavage of IL-1β, GSDMD, and regulation of cytokine transcription[31]. These proposed additional roles of caspase-8 complicate interpretation of disease phenotypes associated with this caspase. It is not clear if death receptor-mediated inflammation and cell death are linked or if the two processes can occur independently. Several reports have shown that TLR, TNFR1, FAS (CD95/APO-1) and TRAILR signalling can induce expression of certain pro-inflammatory cytokines and chemokines in a manner driven at least in part by caspase-8. Intriguingly, this process can be either dependent or independent of the catalytic activity of caspase-8[24,32–35]. Following TNFR1 stimulation, a secondary signalling complex is formed, consisting of FADD oligomerized with caspase-8 and associated with RIPK1, cIAP1/2, TRAF2, and NEMO[14]. This complex functions to activate the NF-κB, JNK, and p38/MAPK signalling pathways, driving cellular proliferation as well as survival and inflammation[24,32,33,36]. Caspase-8 was shown to regulate NF-κB by cleaving and thereby inactivating the negative regulator of cytokine production NEDD4-binding protein 1 (N4BP1)[37]. This pathway is regulated at the death-inducing signalling complex (DISC) by altering caspase-8 processing. The extent of caspase-8 activity, as well as the sites that are cleaved, collectively determine if the cell will survive or die. Full-length pro-caspase-8 dimers can self-cleave and process a specific set of localized DISC-proximal substrates together with the cellular FAS-associated death domain-like interleukin-1β-converting enzyme-inhibitory protein long form (cFLIP_L)[38,39]. cFLIP_L is structurally similar to caspase-8 but lacks catalytic activity. Structural and biochemical studies have shown that cFLIP_L can function as a dynamic (positive or negative) regulator of caspase-8[40,41]. The function of cFLIP_L is dependent on its cytoplasmic concentration and stoichiometry relative to pro-caspase-8[38,39]. cFLIP_L can bind pro-caspase-8 through its death-effector domain (DED), with the cytoplasmic concentration of each of the components determining functional output[38,39,41,42]. In contrast, the splice variant cFLIP short (cFLIP_S), which also lack enzymatic activity, acts to inhibit caspase-8's catalytic and therefore apoptotic activity[40].

During SARS-CoV-2 infection, the release of pro-inflammatory cytokines is associated with severe COVID-19[20,21,43]. Host cell death has been postulated to contribute to this pathology, but genetic in vivo studies are lacking[44]. To model this hyper-inflammation, we infected mice with a clinical SARS-CoV-2 isolate and performed serial in vivo passaging, generating an adapted strain (P21) that induces severe disease[45]. Using this model, we found that deletion of IL-1β, but not IL-18, protects mice from severe pathology, and our previous discoveries challenge the notion that necroptosis and pyroptosis are required for cytokine release and disease severity[22,23]. Although caspases-1 and −11 have direct or indirect roles in processing IL-1β[46–48], we found that mice lacking these inflammatory caspases ($C1^{-/-}/11^{-/-}/12^{-/-}$, short $C1/11/12^{-/-}$) do not display the same level of disease attenuation afforded by the loss of IL-1β[23]. These observations underscore the critical role of gene-targeted animal studies in unravelling the complex pathogenesis of diseases that involve multiple effectors, cell types and organ systems. We have previously shown that animals lacking TNF are significantly protected from severe disease[45]. We now show that, in contrast to other diseases, the effects of TNF in severe SARS-CoV-2 pathology do not need to be potentiated by IFN-γ. TNF can trigger apoptosis or alternatively promote inflammation by inducing NF-κB activation. In this study we uncovered that caspase-8 triggers a pronounced pro-inflammatory transcriptional response and IL-1β release, which is driven by an accumulation of caspase-8 and N4BP1 cleavage in affected tissues. Caspase-8-driven inflammation in vivo is independent of apoptotic or necroptotic cell death but can be reduced by the broad-spectrum caspase inhibitor emricasan. Collectively, these findings suggest new approaches for the prevention and treatment of severe COVID-19, and possibly related viral pathologies.

## Results
### Caspase-8 drives inflammation in severe SARS-CoV-2 induced disease

Our previous work indicated that excessive IL-1β production during SARS-CoV-2 infection is driven by non-canonical (i.e. caspase-1/caspase-11 independent) pathways[23]. Beyond its role in initiating extrinsic apoptosis, caspase-8 has been shown to modulate several pro-inflammatory signalling pathways, including pro-IL-1β processing[49,50]. Since caspase-8 deficiency causes embryonic lethality[51] due to unchecked activation of necroptosis, we used compound mutant mice also deficient in RIPK3 to dissect the role of caspase-8 in SARS-CoV-2 induced disease[52–54]. At 3 days post-infection (3 dpi) with SARS-CoV-2, $C8^{-/-}/R3^{-/-}$ ($C8/R3^{-/-}$) showed less weight loss and a reduction in viral burden compared to WT mice (Fig. 1a, b). Notably, reductions in pathology or viral burden were not observed in animals lacking RIPK3 ($R3^{-/-}$) (Fig. 1a, b), or as previously reported, in mice deficient for $Mlkl$[22]. To determine whether the absence of RIPK3 might be compensated for by RIPK1 signalling in cells from $C8/R3^{-/-}$ animals, we assessed RIPK1 activation in lung tissue by Western blot analysis. Phosphorylated (i.e. activated) RIPK1 was undetectable in both double and single knockout mice (Fig. S1a), indicating that RIPK3 deletion does not alter RIPK1 activation status.

Comparable to COVID-19 outcomes observed in the human population, SARS-CoV-2 P21-driven disease severity in mice escalates with advancing age[45]. Most WT animals older than 10 weeks reach predetermined ethical endpoint by 4-5 dpi (see methods). In contrast, $C8/R3^{-/-}$ animals at 11 weeks of age demonstrated a higher likelihood of surviving SARS-CoV-2 infection (Fig. 1c, d). This was not seen in mice lacking only RIPK3 (Fig. S1b, c). A characteristic of caspase-8 deficient animals that have a compound deficiency for RIPK3 or MLKL is the development of lymphoproliferative (LPR) syndrome with age[54]. A hallmark of LPR syndrome is immune system dysregulation, including the characteristic accumulation of atypical TCRαβ+CD3+B220+CD4−CD8− double-negative T cells in peripheral lymphoid tissues. While this phenotype was absent in our younger cohorts (5–7 weeks), early but mild lymphoproliferative disease was detectable by 10 weeks in compound caspase-8 (and RIPK3)-deficient animals. Although this could subtly influence immune responses in the infected knockout models, the mild nature of the phenotype at this age makes it unlikely to fully account for the infection-associated differences observed.

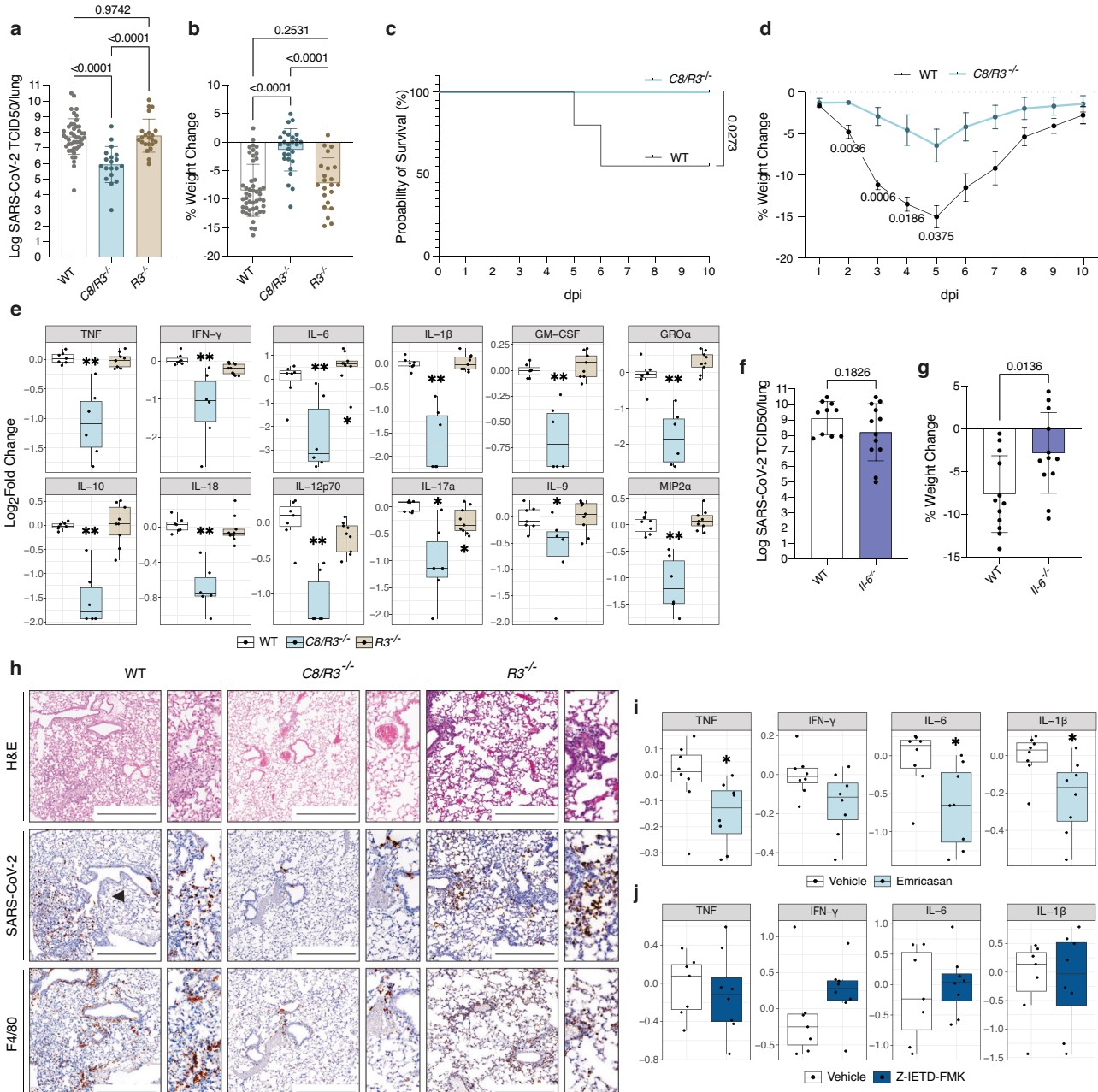

**Fig. 1 | Caspase-8 drives severe disease and cytokine release upon SARS-CoV-2 infection. a,b** C57BL/6 (WT), *caspase-8/Ripk3* knockout (*C8/R3⁻/⁻*) and *Ripk3* knockout (*R3⁻/⁻*) mice were infected with 10⁴ TCID50 of SARS-CoV-2 P21 and monitored at 3 dpi for **(a)** lung viral burden by TCID50 assay and **b** percent weight change compared to initial weight (results are pooled from 4 independent experiments, n(WT) = 42; n(*C8/R3⁻/⁻*) = 19; n(*R3⁻/⁻*) = 22 mice). **(c-d)** 12-week-old WT and *C8/R3⁻/⁻* mice were infected intranasally with 10⁴ TCID50 of SARS-CoV-2 P21 and monitored for **c** percentages of animals that became severely ill (reaching pre-determined ethical endpoint) and required euthanasia and **d** weight loss at different dpi (Mean ± SEM is shown, n(WT) = 18; n(*C8/R3⁻/⁻*) = 7 mice). **e** Levels of cytokines and chemokines measured by ELISA of lung homogenates of WT and knockout animals 3 days post SARS-CoV-2 infection (n(WT) = 7; n(*C8/R3⁻/⁻*) = 6; n(*R3⁻/⁻*) = 9 mice; p-values can be found in the source data file). **f,g** WT and *Il-6* knockout (*Il-6⁻/⁻*) mice were infected with 10⁴ TCID50 of SARS-CoV-2 P21 and monitored at 3 dpi for **f** lung viral burden by TCID50 assay and **g** percent weight change compared to initial weight (n(WT) = 10; n(*Il-6⁻/⁻*) = 12 mice). **h** Representative images of haematoxylin and eosin (H&E) stained and immunohistochemistry (IHC) stained lungs for SARS-CoV-2 nucleocapsid and F4/80 (macrophages) of infected WT and gene targeted animals (images are representative of at least 3 animals per genotype). Scale bar = 500 μm. **i** Levels of cytokines and chemokines measured by ELISA of lung homogenates of WT vehicle or emricasan treated mice at 3 days post intranasal SARS-CoV-2 P21 infection or mock (media only) (n = 8 mice per condition; p-values can be found in the source data file). **j** Levels of cytokines and chemokines measured by ELISA in lung homogenates from WT vehicle and Z-IETD-FMK treated mice at 3 days post SARS-CoV-2 P21 infection or mock (n(Vehicle) = 7; n(IETD) = 8 mice; p-values can be found in the source data file). Data are presented as mean ± SD unless noted otherwise. Boxplots depict the median and interquartile range (IQR). Whiskers extend to the furthest data point within 1.5 times the IQR from each box end. One-way ANOVA with multiple comparisons (**b**) after log10 transformation (**a**), two-way ANOVA with multiple comparisons (**d**) unpaired two-tailed Student's t test after log10 transformation (**f**), unpaired two-tailed Student's t test (**g**), two-sided Wilcoxon rank-sum (**e,i,j**) and two-sided Log-rank Mantel-Cox test (**c**), statistical tests were performed; *P < 0.05 and **P < 0.01. Source data are provided as a Source Data file.

## Caspase-8 drives lung disease and cytokine release upon SARS-CoV-2 infection

To understand the contributions of caspase-8 to the exaggerated pro-inflammatory phenotype, we analysed the cytokine responses in *C8/R3[-/-]* and *R3[-/-]* mice by ELISA during the acute phase of disease. We have previously shown that deficiency of pro-inflammatory caspases-1/−11/−12 did not reduce the levels of IL-1β in SARS-CoV-2 infected mice[23]. In infected lung tissue this key cytokine was only reduced in the absence of caspase-8 (combined with loss of RIPK3) that was not observed upon loss of RIPK3 alone (Fig. 1e). Comparison of cytokines in animals with single deficiency in RIPK3 showed similar responses to WT controls, while *C8/R3[-/-]* mice had reduced levels of a range of pro-inflammatory cytokines, including TNF, IL-6 and GM-CSF (Fig. 1e and S1c). In addition to the critical role of IL-1β, IL-6 levels have also been shown to correlate with poor prognosis in COVID-19 patients[55]. Interestingly, SARS-CoV-2 infected *C8/R3[-/-]* mice also had reduced levels of IL-6. To understand the potential contribution of IL-6, we infected animals lacking this cytokine (*Il-6[-/-]*) and found comparable viral burdens to WT animals (Fig. 1f). However, *Il-6[-/-]* mice showed significantly less weight loss, indicating that this cytokine contributes to the development of severe disease (Fig. 1g). Therefore, the observed reduction of both IL-1β[23] and IL-6 may contribute to the improved health of SARS-CoV-2 infected caspase-8/RIPK3 deficient mice. Collectively, these findings demonstrate that caspase-8 is essential for driving pathogenic inflammation during SARS-CoV-2 infection, and that this likely occurs independent of RIPK3.

To better characterise disease progression in gene-targeted animals following infection with SARS-CoV-2, we compared lung histopathology of WT and knockout mice at 3 dpi. Lung sections were stained with haematoxylin and eosin (H&E) and evaluated by a pathologist. As we have previously reported[45], WT mice exhibited multifocal acute alveolitis, along with perivascular and peribronchiolar lymphocytic infiltration (Fig. 1h). In contrast, *C8/R3[-/-]* mice displayed only mild increase in perivascular and peribronchiolar lymphocytes and neutrophils, and sparse inflammatory cell infiltrates in the subpleural regions. Immunohistochemical staining for myeloperoxidase (MPO), CD3, and F4/80 confirmed an attenuation of immune cell infiltration in caspase-8-deficient animals (Fig. 1h, S1e), consistent with reduced pulmonary inflammation.

## Treatment with emricasan leads to reduced cytokine release during SARS-CoV-2 infection in mice

A caveat arises in interpreting our in vivo data related to the fact that the loss of caspase-8 must always be accompanied by a loss of either RIPK3 or MLKL to prevent embryonic lethality. To better understand the contributions of caspase-8 to severe SARS-CoV-2-driven disease, independently of the necroptosis mediator RIPK3 which itself has been associated with the production of certain cytokines and chemokines beyond its role in necroptosis[54], we treated SARS-CoV-2 infected WT animals with the broad-spectrum caspase inhibitor emricasan (Fig. S1f). Caspase inhibition in the presence of elevated TNF, as observed during infection, is predicted to provoke necroptosis. Accordingly, emricasan-treated SARS-CoV-2 infected animals exhibited significantly greater weight loss, while viral burden remained unchanged (Fig. S1g, h). Notably, emricasan treatment reduced the levels of TNF, IL-6, and IL-1β in lungs of mice at 3 dpi (Fig. 1i). This is consistent with our findings in *C8/R3[-/-]* mice and further indicates a caspase-8-dependent mechanism of cytokine induction. However, emricasan targets multiple caspases[56], limiting interpretation of underlying mechanisms. To improve specificity, we also tested Z-IETD-FMK (Fig. S1i), a more selective caspase-8 inhibitor. Similar to the administration of emricasan, treatment with Z-IETD-FMK increased viral burden and exacerbated weight loss, likely due to uncontrolled necroptosis (Fig. S1i–k) but it did not suppress inflammatory cytokine levels (Fig. 1j). This divergence between emricasan vs. Z-IETD-FMK

likely reflects differential target engagement. Emricasan has been shown to inhibit caspase-8 activity within the caspase-8/cFLIP heterodimer[56], a complex implicated in non-apoptotic, pro-inflammatory signalling. Overall, these findings indicate that caspase-8 may drive inflammation and disease progression during SARS-CoV-2 infection without undergoing full catalytic processing.

## Lung proteomics of SARS-CoV-2 infected animals revealed upregulation of caspase-8 pathway components

To further dissect the role of caspase-8 in pro-inflammatory signalling during SARS-CoV-2 infection, we performed bulk lung proteomics on WT, *C8/R3[-/-]*, and *R3[-/-]* mice at 3 dpi. *R3[-/-]* mice served as a genetic control for the *C8/R3[-/-]* double knockout. We first compared protein expression profiles between SARS-CoV-2 infected and mock control (inoculated intranasally with media only) animals within each genotype. Across all three groups, SARS-CoV-2 infection consistently triggered upregulation of antiviral and immune response pathways and downregulation of metabolic pathways (Fig. S2a). We next investigated genotype-specific responses to infection by comparing proteins differentially expressed upon infection in each genotype (Fig. 2a). Notably, 805 proteins were differentially expressed exclusively in the WT SARS-CoV-2 infected vs. mock control comparison, whereas no unique infection-associated proteins were identified in neither *R3[-/-]* or *C8/R3[-/-]* animals relative to their respective mock controls. A shared subset of 289 proteins was significantly altered in both WT and *R3[-/-]* mice but remained unchanged in *C8/R3[-/-]* animals (Fig. 2a). We hypothesized that this subset of proteins may contribute to caspase-8-dependent disease pathogenesis. Among these proteins, 10 were significantly upregulated in WT compared to *C8/R3[-/-]* mice (Fig. S2b). Functional annotation revealed that these proteins were predominantly involved in metabolic, inflammatory, and iron processes, reinforcing a central role for caspase-8 in mediating immunopathology during SARS-CoV-2 infection (Fig. 2b). Of note, dysregulation of iron metabolism has been linked to increased pathology during acute SARS-CoV-2 disease[57], as well as long COVID[58], but its link to caspase-8 signalling warrants further investigation.

A direct comparison of infected *C8/R3[-/-]* vs. WT proteomes highlighted pronounced downregulation of acute-phase proteins, inflammatory mediators, components of iron metabolism and plasma lipoproteins in the former (Fig. 2c, d). These changes mirror clinical observations linking altered plasma lipid and fatty acid metabolism with COVID-19 severity in humans[59]. Conversely, SARS-CoV-2 infected *C8/R3[-/-]* animals showed increased expression of pathways related to humoral immune responses, protein stability, phagocytosis, and complement activation (Fig. 2c, d). Compared to SARS-CoV-2 infected *R3[-/-]*, *C8/R3[-/-]* animals showed increased expression of proteins related to phagocytic, complement and humoral responses, while cytokine production and NF-kB signalling were downregulated (Fig. 2e). These results suggest a significant role for caspase-8 in regulation of inflammation and immune responses during SARS-CoV-2 infection. Importantly, SARS-CoV-2 infected *R3[-/-]* vs. WT comparisons revealed distinct signatures compared to SARS-CoV-2 infected *C8/R3[-/-]* vs. infected WT comparisons, indicating that caspase-8 exerts effects independently of RIPK3 during infection (Fig. 2f).

## Spatial transcriptomic analysis reveals altered tissue niches in the absence of caspase-8

To better understand the impact of caspase-8 deletion during SARS-CoV-2 infection in vivo, we performed spatial transcriptomic analysis on lung tissue from WT infected and mock, as well as infected *R3[-/-]* and *C8/R3[-/-]* mice at 3 dpi. Using LungMAP reference data, we identified 15 distinct cell clusters. Eleven clusters corresponded to known cell types, while four additional clusters were defined by strong expression of a single marker gene (*Ifnb1*, *Fosb*, *Ptx3* and *Ptgs2*), in several cell types (Fig. S2c–e). To account for experimental variability, we focused our

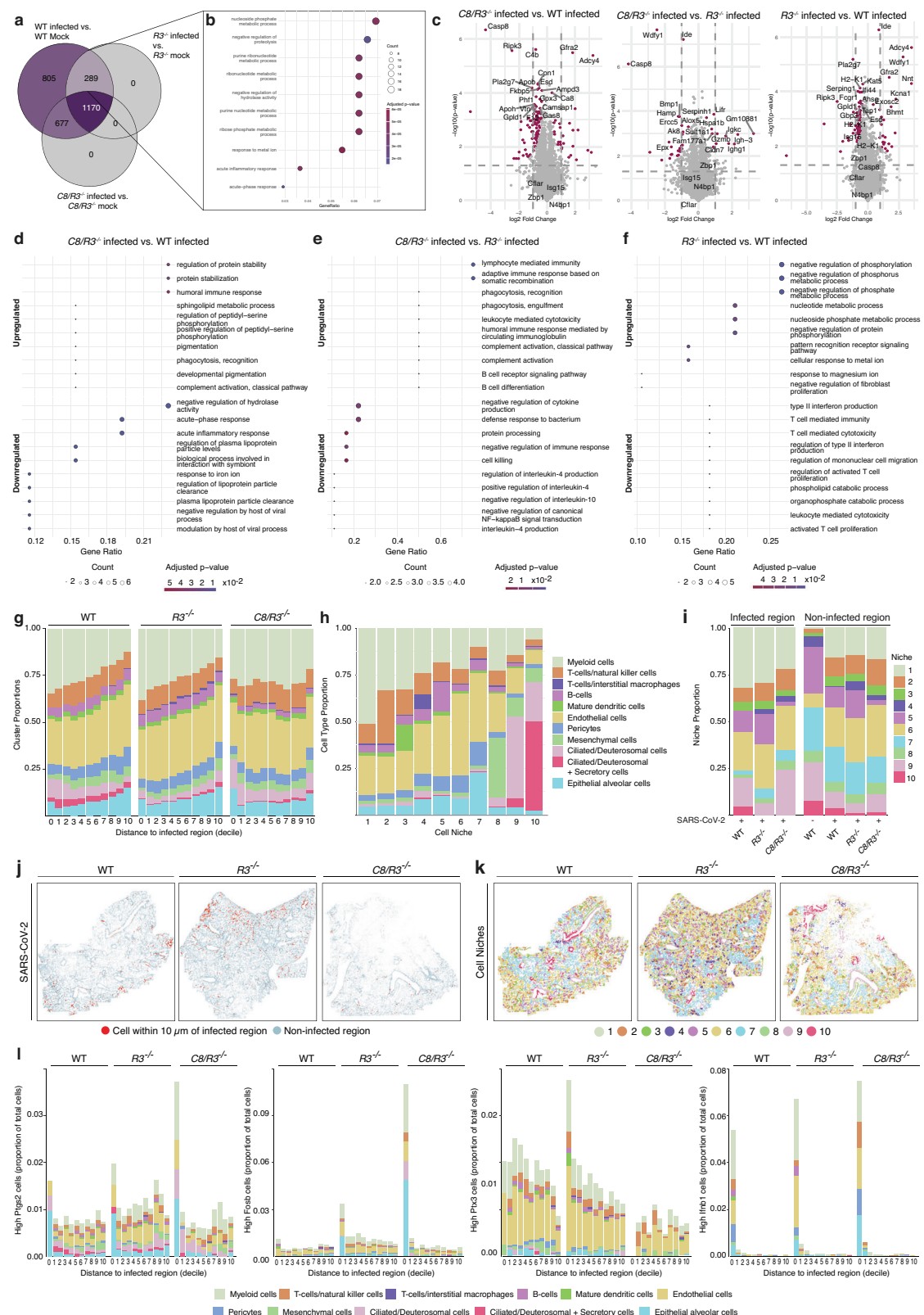

comparisons on infected vs. uninfected regions within a section, which are largely internally controlled. Interestingly, WT and *R3⁻ᐟ⁻* animals revealed enhanced myeloid cell infiltration on and near infected sites, compared with *C8/R3⁻ᐟ⁻* mice (Fig. 2g).

To explore spatial context, each cell was assigned to one of ten spatial "niches" based on the composition of its 30 nearest neighbours (Fig. 2h), focusing on comparing infected and non-infected

areas across genotypes. Niches 9 and 10 represent areas around the bronchi, which are more prominent in lungs of WT and *C8/R3⁻ᐟ⁻* than in the *R3⁻ᐟ⁻* mice, likely due to sectioning effects (Fig. 2h–k and Fig. S2f). Infection is more localised to these regions in the *C8/R3⁻ᐟ⁻* mouse lung than in the WT and *R3⁻ᐟ⁻* controls, indicating reduced spread of viral particles beyond the main airways. Overall, infection appears to mobilise myeloid cells in WT and *R3⁻ᐟ⁻* infected lungs, but

**Fig. 2 | The absence of caspase-8 affects immune related pathways during infection with SARS-CoV-2. a–f** C57BL/6 (WT), *caspase-8/Ripk3* knockout (*C8/R3⁻ᐟ⁻*) and *Ripk3* knockout (*R3⁻ᐟ⁻*) mice were inoculated intranasally with either mock (media only) or SARS-CoV-2 P21 and lungs were taken at 3 dpi for bulk proteomics analysis (n(WT mock) = 4; n(WT infected) = 4; n(*C8/R3⁻ᐟ⁻* mock) = 4; n(*C8/R3⁻ᐟ⁻* infected) = 4; (*R3⁻ᐟ⁻* mock) = 4; n(*R3⁻ᐟ⁻* infected) = 4). **a** Venn Diagram comparing protein expression profiles between infected and mock animals within each genotype. **b** Pathway enrichment was performed on the 289 proteins common to the infected WT vs. WT and infected *R3⁻ᐟ⁻* vs. mock *R3⁻ᐟ⁻* comparisons, which are not present in infected *C8/R3⁻ᐟ⁻* vs. mock *C8/R3⁻ᐟ⁻* comparison **c** Volcano plot of proteins regulated in mouse lungs at 3 dpi in diverse comparisons in shown. Purple: significantly differentially expressed proteins. **d–f** Dot plots showing the top 10 enriched GO terms among significantly upregulated (top panel) and downregulated (bottom panel) genes, selected based on gene ratio and adjusted p-value. **g–l** C57BL/6 (WT), *caspase-8/Ripk3* knockout (*C8/R3⁻ᐟ⁻*) and *Ripk3* knockout (*R3⁻ᐟ⁻*) mice were inoculated intranasally with either mock (media only) or SARS-CoV-2 P21 and lungs were taken at 3 dpi for spatial transcriptomic analysis using MERSCOPE. **g** Cell

cluster proportions plotted across deciles of increasing distance from SARS-CoV-2–infected regions (cell types are colour coded as in (**h**)). Deciles 1-10 represent cells outside the infected regions. "Decile 0" is used to represent cells inside the infected regions. **h** Proportions of cell types within ten spatially-defined "niches", based on 30 nearest neighbours. **i** Niche distribution in infected versus non-infected regions, stratified by genotype. **j** Spatial maps of SARS-CoV-2–positive (red) and non-infected (grey) regions. Red points mark cells located within 10 μm of an infected region. **k** Niche maps showing spatial distribution of each of the ten niches across genotypes. **l** Cell cluster proportions expressing *Fosb, Ifnb1, Ptgs2, Ptx3* were plotted across deciles of increasing distance from SARS-CoV-2-infected regions. Source data are provided as a Source Data file. Pathway enrichment was assessed using clusterProfiler, with multiple testing correction by the Benjamini-Hochberg method (b, d, e, f). Differential expression was assessed using the limma/voom pipeline with moderated t-statistics and empirical Bayes shrinkage of variance estimates; p values were adjusted for multiple testing using the Benjamini-Hochberg method (c).

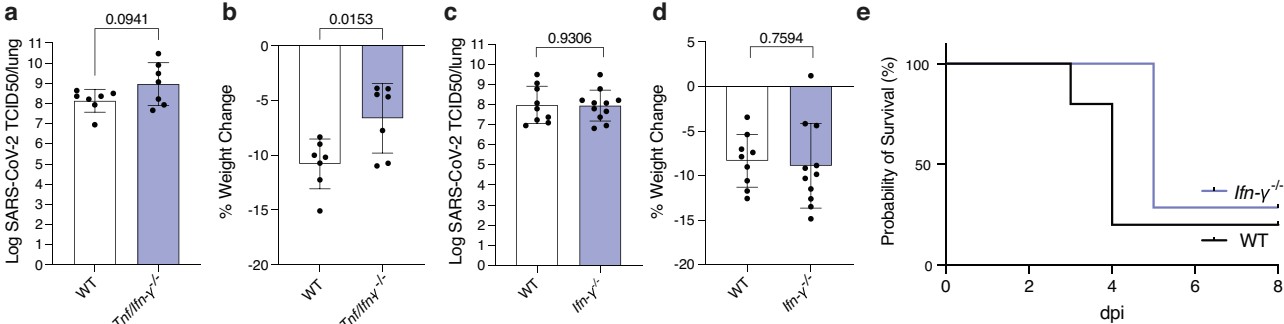

**Fig. 3 | Caspase-8 contributes to SARS-CoV-2 induced severe disease independently of IFN-γ. a,b** WT and *Tnf/Ifn-γ* knockout (*Tnf/Ifn-γ⁻ᐟ⁻*) mice were infected with 10⁴ TCID50 of SARS-CoV-2 P21 and monitored at 3 dpi for **a** lung viral burden by TCID50 assay and **b** percent weight change compared to initial weight (n(WT) = 7, n(*Tnf/Ifn-γ⁻ᐟ⁻*) = 7 mice). **c,d** WT and *Ifn-γ* knockout (*Ifn-γ⁻ᐟ⁻*) mice were infected with 10⁴ TCID50 of SARS-CoV-2 P21 and monitored at 3 dpi for **c** lung viral burden by TCID50 assay and **d** percent weight change compared to initial weight (results are pooled from 2 independent experiments, n(WT) = 9; n(*Ifn-γ⁻ᐟ⁻*) = 11). **e** 6-month-old

WT and *Ifn-γ⁻ᐟ⁻* animals were infected intranasally with 10⁴ TCID50 of SARS-CoV-2 P21 and monitored for the proportion of mice that became severely ill (reaching predetermined ethical endpoint, n(WT) = 5; n(*Ifn-γ⁻ᐟ⁻*) = 7; p = 0.0661). Data are presented as mean ± SD. Unpaired two-tailed Student's t test after log10 transformation (a, c), unpaired two-tailed Student's t test (b, d), and two-sided Log-rank Mantel-Cox test (e), statistical tests were performed; *P < 0.05. Source data are provided as a Source Data file.

not in *C8/R3⁻ᐟ⁻* mice, and there is a relative lack of B cells in *C8/R3⁻ᐟ⁻* infected lungs, which might contribute to the observed phenotype (Fig. 2g–i).

To dissect the functional significance of the emergent *Ifnb1, Fosb, Ptx3* and *Ptgs2* clusters, we quantified the proportion of positive cells by genotype and distance from infected regions (Fig. 2l). This revealed a striking enrichment of *Ptgs2⁺* and *Fosb⁺* cells within infected regions of lungs of *C8/R3⁻ᐟ⁻* mice, compared to WT or *R3⁻ᐟ⁻* controls. Although cyclo-oxygenase-2 (COX-2/PTGS2) is classically viewed as pro-inflammatory, epithelial COX-2 activity can accelerate alveolar restitution and curtail fibrotic scarring[60,61]. Similarly, *Fosb⁺* cells, previously reported to decrease during the acute phase of SARS-CoV-2 pneumonia and rebound during convalescence, support the notion that caspase-8 deficiency could preserve a regenerative state[62,63]. In contrast, the frequency of *Ptx3⁺* cells were significantly reduced across the whole lungs of *C8/R3⁻ᐟ⁻* mice. Because PTX3 levels correlate with endothelial activation, immunothrombosis and mortality in severe COVID-19[64], their reduction suggests dampened vascular injury and improved barrier integrity in the absence of caspase-8. Finally, *C8/R3⁻ᐟ⁻* lungs showed a modest but spatially confined augmentation of *Ifnb1* (Fig. 2l). Such a shift from destructive inflammation toward coordinated tissue repair may underlie the improved outcomes observed in *C8/R3⁻ᐟ⁻* mice and highlights caspase-8 as a potential therapeutic node for modulating host responses to SARS-CoV-2.

## Caspase-8 driven severe disease is dependent on TNF but does not require IFN-γ

We and others have previously shown that TNF contributes to disease severity upon SARS-CoV-2 infection[43,45,65,66]. Notably, reports have highlighted a synergistic interplay between IFN-γ and TNF in driving caspase-8 activation across various disease contexts[43,67]. IFN-γ has been associated with caspase-8 upregulation and activation in the context of cancer[68], and emerging literature proposes downstream activation of caspase-8 following IFN-γ priming during infection with certain pathogens, including SARS-CoV-2[34,69]. To test whether these cytokines act synergistically in vivo, we infected *Tnf* and *Ifn-γ* compound gene knockout mice (*Tnf⁻ᐟ⁻/Ifn-γ⁻ᐟ⁻*, short *Tnf/Ifn-γ⁻ᐟ⁻*). Viral loads were similar compared to WT animals, however *Tnf/Ifn-γ⁻ᐟ⁻* showed reduced weight loss (Fig. 3a, b), which was not substantially different from animals lacking only TNF, as we had previously published[45]. Mice lacking IFN-γ showed similar disease outcomes compared to infected WT control mice (Fig. 3c, d). This indicates that although IFN-γ levels are elevated during SARS-CoV-2 infection, they do not appear to be a critical driver of pathogenesis. Additionally, unlike TNF[45], we found no indication that IFN-γ influences disease severity in the context of age, as 6-month-old *Ifn-γ⁻ᐟ⁻* mice succumbed to SARS-CoV-2 infection with kinetics similar to WT animals (Fig. 3e). These findings underscore TNF's central role in determining disease severity during SARS-CoV-2 infection and reveal that elevated IFN-γ does not appear to significantly impact disease progression in mice.

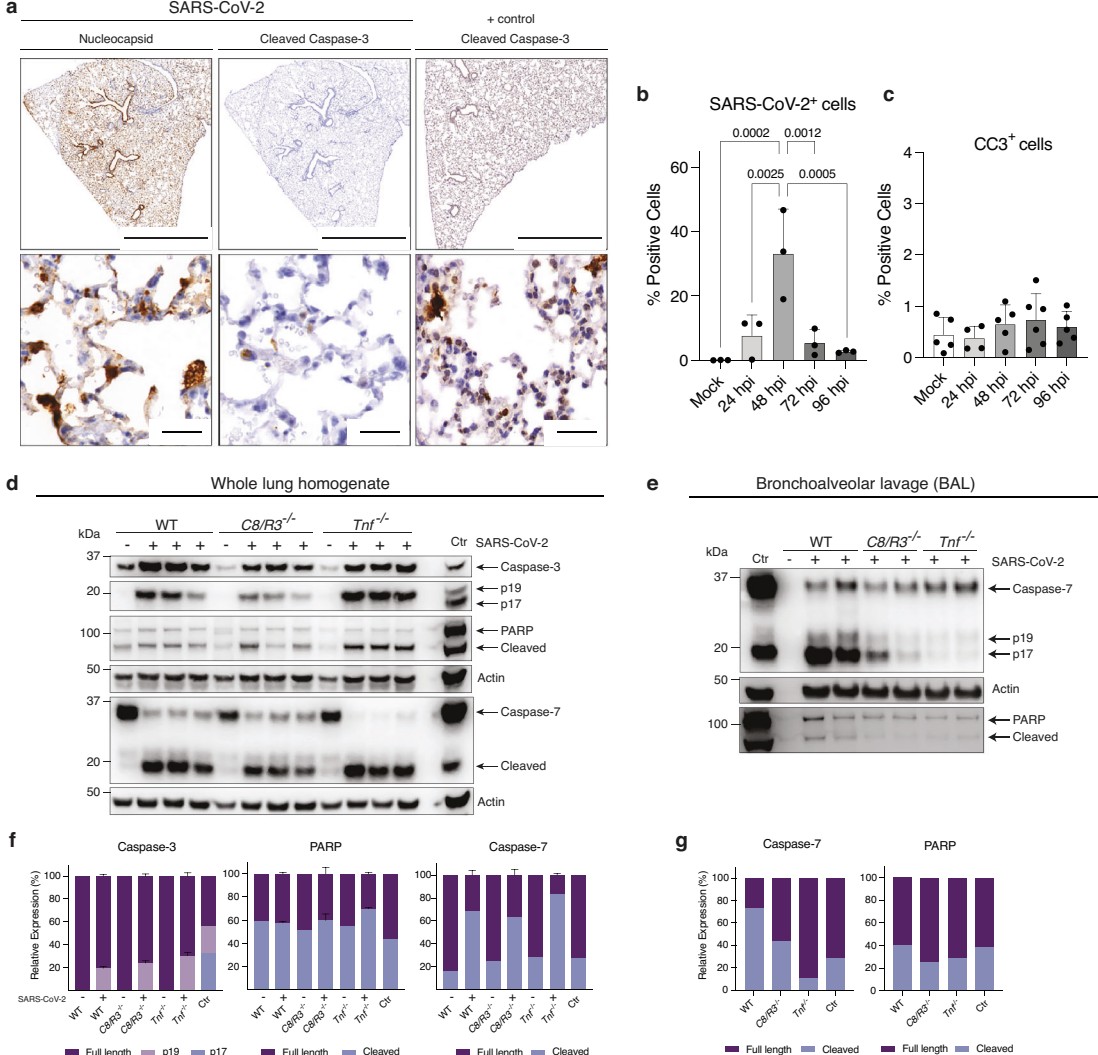

**Fig. 4 | Caspase-8 driven SARS-CoV-2 infection induced disease is not due to overt apoptosis. a** Representative images of immunohistochemistry (IHC) stained lungs showing SARS-CoV-2 nucleocapsid and cleaved (i.e. activated) caspase-3 (CC3). Aged (12-week-olds) WT mice were infected with $10^4$ TCID50 of SARS-CoV-2 P21 and lungs were collected at 3 days post-infection (dpi) for histological analysis. A positive control was employed by taking lungs of $p50^{-/-}$ mice infected with lymphocytic choriomeningitis virus (LCMV) and stained for CC3. Histological images are representative of at least 3 mice per genotype. Scale bar (top) = 2000 μm. Scale bar (bottom) = 50 μm. **b,c** IHC staining was quantified at different time points post SARS-Cov-2 infection. Percentages of positive cells were calculated following counting of cells stained for **b** SARS-CoV-2 nucleocapsid (n = 3 animals per time-point) or **c** CC3 (n(mock) = 4; n(24 h) = 4; n(48 h) = 5; n(72 h) = 6; n(96 h) = 5 animals

per timepoint). **d,e** Western blot analysis of **d** whole lungs or **e** bronchoalveolar lavage (BAL) from SARS-CoV-2 P21 infected (+) or mock controls (-) WT, $C8/R3^{-/-}$ and $Tnf^{-/-}$ mice at 3 dpi ($10^4$ TCID50). Samples were probed for full-length caspase-3, cleaved (i.e. activated) caspase-3, PARP and caspase-7 (including cleaved, i.e. activated). Probing for actin is shown as a loading control. Ctr.: bone marrow-derived macrophages treated with TNF and SMAC mimetics. **f,g** The relative expression of the proteins of interest indicated were quantified and normalized to corresponding levels of actin. Each well represents a biological replicate (lung homogenate of one animal). Data of infected animals (n = 3) are presented as mean ± SD. Western blots are representative of 3 independent experiments. (b,c) One-way ANOVA with multiple comparisons statistical tests were performed. Source data are provided as a Source Data file.

## Apoptosis-independent functions of caspase-8 contribute to SARS-CoV-2-induced disease

To understand if caspase-8 promotes cell death in severe SARS-CoV-2 disease, we first determined if apoptosis was a major feature in aged animals infected with SARS-CoV-2. We performed histological analysis of lungs at various time points post infection. Aged animals are highly susceptible to SARS-CoV-2 P21 infection and exhibit increased disease and mortality compared to younger WT mice[45]. At peak of infection (3 dpi), IHC staining showed SARS-CoV-2 had widely disseminated throughout the lungs, with positive staining in alveolar pneumocytes, the bronchiolar epithelium, and macrophages (Fig. 4a). However, while some apoptosis was observed in the lungs this was not prominent, as indicated by a small number of dispersed cells that showed activated,

i.e. cleaved caspase-3 (CC3) (Fig. 4a). To confirm that the antibody does indeed detect CC3, we stained lung tissue from mice that are highly susceptible to inflammation associated apoptosis (p50 deficient animals) following lymphocytic choriomeningitis virus infection (LCMV)[70], (Fig. 4a). Quantitative assessment of histology images did not show a significant increase in CC3 in SARS-CoV-2 infected animals compared to mock control mice. Whilst SARS-CoV-2 nucleocapsid levels vary during the course of infection, CC3 levels remained constant and did not correlate with viral levels (Fig. 4b, c).

To obtain a more sensitive readout of apoptotic signalling, we performed western blot analysis on whole lung lysates from mock and SARS-CoV-2 infected WT, $C8/R3^{-/-}$ and $Tnf^{-/-}$ mice. TNF is a known activator of caspase-8 signalling and we have previously shown that $Tnf^{-/-}$

animals showed reduced weight loss in our model[45]. While mock control animals showed basal levels of CC3, SARS-CoV-2 infection induced increased caspase-3 processing (Fig. 4d). Notably, this increase was also observed in infected *C8/R3*[-/-] and *Tnf*[-/-] mice. However, comparison with bone marrow-derived macrophages treated with TNF and SMAC mimetics (inhibiting XIAP, cIAP1/2 and thereby activating caspases) as a positive control[71], revealed that caspase-3 was not fully processed to its p17 form during infection in vivo. Instead, SARS-CoV-2 infection primarily led to activation and cleavage of caspase-7 (Fig. 4d–f). To confirm that apoptosis was occurring, we assessed PARP cleavage as a surrogate marker. Infection induced PARP cleavage across all genotypes (Fig. 4d–f). These findings indicate that apoptosis upon infection with SARS-CoV-2 in vivo proceeds predominantly through caspase-7 rather than caspase-3 and is independent of TNF and caspase-8. Furthermore, levels of apoptosis and marker cleavage products were similar across WT, *C8/R3*[-/-] and *Tnf*[-/-] mice, suggesting that the reduced disease severity observed in the latter groups is not due to altered apoptotic signalling. To more precisely examine the role of apoptosis in different cell types, we probed for CC7 in the bronchoalveolar lavage (BAL) of infected animals. Although protein levels in mock control animals were low due to the absence of immune cell infiltrates in the lung, stalled CC3 cleavage was also observed (Fig. S3a). Increased levels of CC7 upon SARS-CoV-2 infection were evident in WT, *C8/R3*[-/-] and *Tnf*[-/-] mice. CC7 and PARP cleavage were less pronounced in *C8/R3*[-/-] and *Tnf*[-/-] animals, but quantification with actin normalization revealed subtle differences between WT and gene targeted animals (Fig. 4e, g).

Caspase-8 is essential for all extrinsic apoptosis[72]. Therefore, the detection of apoptosis in *C8/Ripk3*[-/-] mice suggests activation of the intrinsic apoptotic pathway during infection with SARS-CoV-2. Although the levels of full-length pro-caspase-9 were increased in *C8/R3*[-/-] animals, SARS-CoV-2 infection did not lead to an increase in caspase-9 cleavage, a sign of its activation, in the lungs (Fig. S3b). We observed an increase in the pro-apoptotic BH3-only protein BIM[73,74] in *C8/Ripk3*[-/-] mice, perhaps indicating a potential compensatory mechanism for engagement of the intrinsic apoptotic pathway (Fig. S3b). Further characterization of proteins involved in the intrinsic apoptotic pathway showed no differences in the levels of pro-apoptotic BID or anti-apoptotic MCL-1 between WT and *C8/R3*[-/-] animals upon infection but did reveal increased levels of anti-apoptotic BCL-XL in mock control *C8/R3*[-/-] mice, and this was slightly diminished upon infection with SARS-CoV-2 (Fig. S3c). Collectively, these findings demonstrate that while SARS-CoV-2 infection is associated with some induction of apoptosis in the lungs, this is not widespread according to histological examination. Importantly, apoptosis still occurs in mice lacking caspase-8 (and RIPK3), or TNF, which are both protected from severe SARS-CoV-2 induced disease. This is consistent with the notion that apoptosis is not a major driver of pathology in this context. Instead, our findings indicate that caspase-8 likely drives SARS-CoV-2 induced illness by increasing the production of pro-inflammatory cytokines.

### Absence of additional pro-inflammatory caspases does not further ameliorate SARS-CoV-2 disease beyond the loss of caspase-8 alone

In recent years, functional redundancy, as well as compensatory or additive interactions among pro-inflammatory caspases, have been increasingly recognized as key modulators of disease outcomes in infectious and inflammatory conditions[75,76]. To interrogate if such interconnectivity and/or functional overlap of caspase signalling may also contribute to immune responses during SARS-CoV-2 disease, beyond the role of caspase-8, we infected mice lacking the initiator caspases-1, −11, −12, as well as caspase-8 and the necroptosis effector RIPK3 (*C1*[-/-]/*11*[-/-]/*12*[-/-]/*8*[-/-]/*R3*[-/-] short *C1/11/12/8/R3*[-/-] mice) with SARS-CoV-2 P21. At 3 dpi, these mice showed a significant reduction in viral burden,

as well as almost complete protection from weight loss compared to infected WT mice (Fig. 5a, b). However, the level of protection was not greater than that observed in mice lacking only caspase-8 and RIPK3 (see Fig. 1a-b). To gain further insight into the specific role of caspase-8 in this context we infected *C1*[-/-]/*11*[-/-]/*12*[-/-]/*R3*[-/-] (*C1/11/12/R3*[-/-]) animals with SARS-CoV-2. These animals showed similar viral burden, but increased weight loss compared to infected WT mice (Fig. 5a, b). We next compared cytokine and chemokine profiles between infected *C1/11/12/8/R3*[-/-] and *C1/11/12/R3*[-/-] mice. At 3 dpi, mice expressing caspase-8 but lacking caspases-1/11/12 (*C1/11/12/R3*[-/-]) displayed minor differences in only a small selection of the cytokines examined (IL-4, MCP-3, and MIP2α) compared to WT animals (Fig. 5c). In contrast, lung homogenates from infected *C1/11/12/8/R3*[-/-] mice exhibited significant reductions in a wide array of factors, including IFN-γ, IL-1β, IL-10, IL-18 and TNF (Fig. 5c and Fig. S4a). These findings illustrate the dominant role of caspase-8 with no major contributions from caspases-1/11/12 in driving the cytokine storm during SARS-CoV-2 infection. Consistent with this caspase-8-driven inflammatory phenotype, infection of 11-week-old animals with SARS-CoV-2 showed that while WT controls exhibited pronounced weight loss reaching predetermined ethical endpoint necessitating euthanasia, *C1/11/12/8/R3*[-/-] mice were significantly protected and were more likely to survive SARS-CoV-2 infection (Fig. 5d, e), akin to *C8/R3*[-/-] animals (Fig. 1e).

Histological analysis showed interstitial pneumonia, multifocal acute alveolitis and moderate to severe multifocal perivasculitis in SARS-CoV-2 infected WT animals. Interestingly, this was significantly diminished in infected *C1/11/12/8/R3*[-/-] mice (Fig. 5f and Fig. S4b). Quantification of disease hallmarks in H&E-stained sections by an American board-certified pathologist revealed that *C1/11/12/8/R3*[-/-] mice are significantly protected from lung damage (Fig. S4c), similar to *C8/R3*[-/-] animals (Fig. 1h). Collectively, these findings illustrate the dominant role of caspase-8, but no marked contributions by caspases-1/11/12, in driving the cytokine storm, and hence disease severity, during SARS-CoV-2 infection.

### Transcriptomic profiling reveals that caspase-8 driven inflammation during SARS-CoV-2 infection is independent of inflammatory caspases-1 and −11

We next performed RNA sequencing (RNA-seq) on bulk lung homogenates and compared SARS-CoV-2 infected vs. mock WT, *C1/11/12/8/R3*[-/-] and *C1/11/12/R3*[-/-] animals at 3 dpi to identify significantly differentially expressed genes [DEGs, false discovery rate (FDR)-adjusted P value < 0.05 and absolute log2 fold- change > 1] between different genotypes (Fig. 5g, h and Fig. S4d, e). This approach was designed to uncover transcriptional signatures uniquely dependent on caspase-8, while controlling for potential crosstalk with inflammatory caspases (caspases-1, −11 and −12). Compared to WT animals, *C1/11/12/8/R3*[-/-] mice exhibited reduced expression of genes associated with pathogen recognition, cytokine production, and responses to lipopolysaccharides. In contrast, pathways related to erythrocyte function and cellular metabolism were upregulated (Fig. 5g). Notably, direct comparison of infection in *C1/11/12/8/R3*[-/-] and *C1/11/12/R3*[-/-] mice revealed that the additional deletion of caspase-8 resulted in a distinct upregulation of genes linked to pathogen killing and apoptotic processes. Furthermore, loss of caspase-8 led to a marked reduction in IL-1 signalling, STAT pathway activation, and acute-phase responses relative to caspase-8−expressing controls, reinforcing the central role of caspase-8 in driving inflammatory transcriptional programs independently of caspase-1/−11/−12 activity (Fig. 5h).

### SARS-CoV-2 infection causes increased levels of unprocessed caspase-8 in stromal and immune cells in the lung

To understand how caspase-8 drives inflammation during severe SARS-CoV-2 disease, despite the absence of inducing substantial apoptosis, we performed IHC to examine caspase-8 expression in the

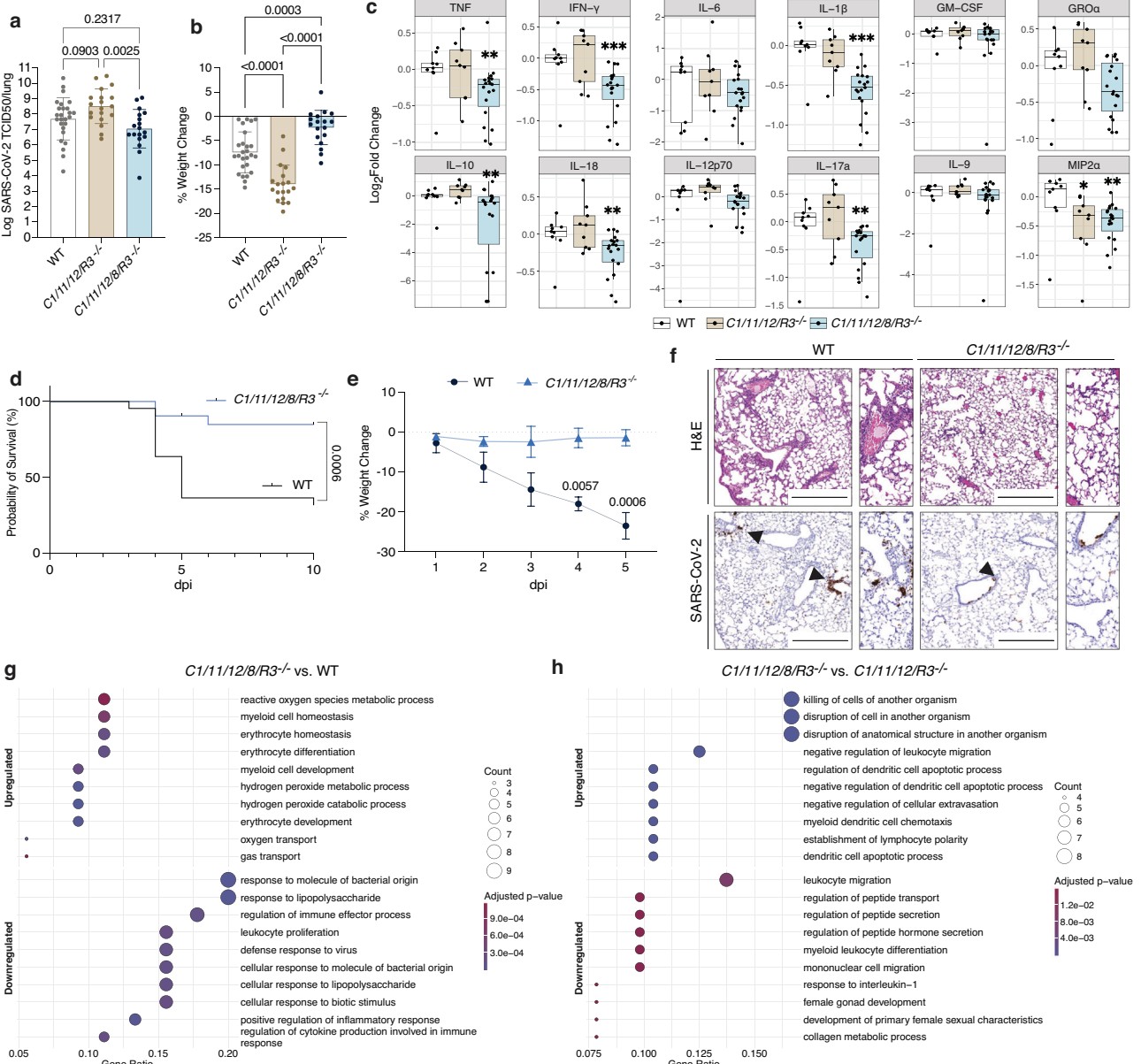

**Fig. 5 | Caspase-8 drives inflammation and SARS-CoV-2 viral dissemination independently of Caspases-1/-11/-12 and RIPK3. a,b** WT, caspase-1, -11, -12, Ripk3 knockout (*C1/11/12/R3⁻/⁻*) and caspase-1, -11, -12, -8, Ripk3 quintuple knockout (*C1/11/12/8/R3⁻/⁻*) mice were infected with 10⁴ TCID50 of SARS-CoV-2 P21 and monitored at 3 days post-infection (dpi) for **a** lung viral burden by TCID50 assay and **b** percent weight change compared to initial weight (results are pooled from 3 independent experiments; n(WT) = 26; n(*C1/11/12/8/R3⁻/⁻*) = 19; n(*C1/11/12/R3⁻/⁻*) = 18 mice). **c** Levels of cytokines and chemokines measured by in lung homogenates 3 dpi (n(WT) = 9; n(*C1/11/12/8/R3⁻/⁻*) = 9; n(*C1/11/12/R3⁻/⁻*) = 19 mice per genotype; p-values can be found in the source data file). Boxplots depict the median and interquartile range (IQR). Whiskers extend to the furthest data point within 1.5 times the IQR from each box end. **d,e** 12-week-old WT and *C1/11/12/8/R3⁻/⁻* mice were infected intranasally with 10⁴ TCID50 of SARS-CoV-2 P21 and monitored for **d** percentages of animals that became severely ill (reaching predetermined ethical endpoint (n(WT) = 24; n(*C1/11/12/8/R3⁻/⁻*) = 25 animals) and **e** weight loss (compared to starting weight) at different dpi (n(WT) = 4; n(*C1/11/12/8/R3⁻/⁻*) = 3 animals). **f** Representative images of haematoxylin and eosin (H&E) and immunohistochemistry (IHC) stained lungs for SARS-CoV-2 nucleocapsid of infected WT and *C1/11/12/8/R3⁻/⁻* mice (images are

representative of at least 3 mice per genotype). Black arrows point to examples of SARS-CoV-2 positive cells. Scale bar = 500 μm. **g,h** Pathway enrichment analysis of significantly differentially expressed genes identified from comparisons of SARS-CoV-2 infected *C1/11/12/8/R3⁻/⁻* mice vs. infected WT mice and SARS-CoV-2 infected *C1/11/12/8/R3⁻/⁻* mice vs. infected *C1/11/12/R3⁻/⁻* mice using Hallmark gene sets. Negative log₁₀ FDR-adjusted P values associated with each pathway are plotted; dot sizes correspond to the proportion of all genes from that pathway that were found to be significantly differentially expressed in each comparison (Gene Ratio). Pathway enrichment was assessed using clusterProfiler, with multiple testing correction by the Benjamini-Hochberg method (n(WT mock) = 4; n(WT infected) = 6; n(*C1/11/12/R3⁻/⁻* mock) = 4; n(*C1/11/12/R3⁻/⁻* infected) = 6; n(*C1/11/12/8/R3⁻/⁻* mock) = 4; n(*C1/11/12/8/R3⁻/⁻* infected) = 6). Data are presented as mean ± SD. One-way ANOVA with multiple comparisons after log₁₀ transformation (**a**), One-way ANOVA with multiple comparisons (**b**), two-sided Wilcoxon rank-sum (**c**), two-sided Log-rank Mantel-Cox test (**d**) and two-way ANOVA with multiple comparisons (**e**) statistical tests were performed; *P < 0.05, **P < 0.01 and ***P < 0.001. Source data are provided as a Source Data file.

lungs from SARS-CoV-2 infected WT mice. While lungs from uninfected mice showed some detectable full-length caspase-8 staining (Fig. 6a), SARS-CoV-2 infected WT mice at 3 dpi showed a significant increase of caspase-8 protein (Fig. 6a) compared to lungs from WT mock controls and from infected *C1/11/12/8/R3$^{-/-}$* mice (the latter used as a negative control for caspase-8 staining) (Fig. S5a). Caspase-8 staining was most prevalent in regions with clear SARS-CoV-2 nucleocapsid staining and areas of lung pathology, although caspase-8 did not always co-localize with viral particles (Fig. 6b). No caspase-3 cleavage (i.e. caspase-3 activation) was evident in these areas (Fig. 6b). In SARS-CoV-2 infected animals, caspase-8 expression was most prominent in bronchiolar and alveolar epithelial cells, as well as in immune cells, with strong staining observed in both macrophages and pneumocytes (Fig. 6b, Fig. S5b). Notably, caspase-8 levels increased as early as 1 dpi. Aged animals, which develop more severe disease, exhibited pronounced caspase-8 staining, supporting its involvement in disease pathogenesis (Fig. S5c). Importantly, elevated caspase-8 levels persisted even after viral clearance, suggesting a potential role in sustaining post-viral inflammation. While murine SARS-CoV-2 infection resolves at 7 dpi[45], after one month levels of caspase-8 were still elevated in the lungs, both in the epithelia, as well as in immune cell infiltrates (Fig. 6c).

Published reports indicate that, while full processing of caspase-8 is necessary for downstream activation of caspases-3/−7 and initiation of apoptosis, the pro-inflammatory function of caspase-8 does not require its full processing, but is dependent on its oligomerization[24]. Western blot analysis of whole lung tissue from infected WT mice confirmed high levels of caspase-8 protein, as well as increased levels of p41/43, the first cleavage step in the activation cascade of caspase-8, a known requirement for oligomerization and pro-inflammatory signalling[24] (Fig. 6d). Some cleavage of caspase-8 to its apoptotic form (p18 oligomer) could also be detected (Fig. 6d). However, this cannot be the dominant driver of caspase-3 processing as levels of CC3 are not reduced in tissues from mice lacking caspase-8 (and RIPK3) (see Fig. 3d–f). It therefore appears unlikely that the absence of caspase-8 protects from lethal SARS-CoV-2 disease due to abrogation of its role in apoptosis.

Caspase-8 signalling is controlled by its catalytically inactive relative, cFLIP$_L$, which oligomerizes with caspase-8 changing its function based on the relative cytoplasmic concentrations of these two proteins[38,39]. In whole lungs, Western blot analysis showed an increase in cFLIP$_L$ levels upon SARS-CoV-2 infection in both WT and *C8/Ripk3$^{-/-}$* mice (Fig. 6d). We further performed Western blot analysis of BAL from infected mice to examine caspase-8 signalling in infiltrating immune cells. Upon SARS-CoV-2 infection there was also a prominent accumulation of full-length caspase-8, and some cleavage of caspase-8 to the p41/43 and p18 fragments in hematopoietic cells in the lungs, while cFLIP$_L$ levels remained similar to those found in uninfected mice (Fig. 6d). This indicates that in hematopoietic cells in the BAL the more prominent caspase-8 p18 fragment may reflect a consequence of an altered caspase-8:cFLIP$_L$ stoichiometry.

## SARS-CoV-2 infection leads to an increase in caspase-8 in lung tissue explants from humans

To evaluate the relevance of our findings in a human context, we infected lung explant tissue derived from patient biopsies[77] with SARS-CoV-2 and performed histological analysis for caspase-8, CC3, and SARS-CoV-2 nucleocapsid protein. Uninfected tissue exhibited low levels of caspase-8, while some CC3 staining was present, likely reflecting spontaneous apoptosis during ex vivo culture (Fig. 6e). Following infection, caspase-8 protein levels increased, particularly in regions adjacent to nucleocapsid-positive cells, whereas CC3 (activated caspase-3) levels remained comparable to uninfected controls, suggesting that caspase-8 activation was not accompanied by widespread apoptotic cell death (Fig. 6e).

To further investigate the pathways activated by caspase-8, we performed spatial transcriptomic profiling on uninfected and SARS-CoV-2 infected lung explants. Multiple fields of view consistently showed only modest transcriptional upregulation of caspase-8 (Fig. 6f). This observation aligns with our previously published lung transcriptomic data from SARS-CoV-2 P21-infected mice, where caspase-8 mRNA levels remained unchanged compared to control animals[45]. This indicates that caspase-8 may be up-regulated post-transcriptionally. Despite the lack of robust transcriptional induction of *CASP8*, we observed increased expression of multiple genes involved in caspase-8 regulated signalling, including cFLIP (*cflar*), *Il-1β*, *Tnf* and death receptors such as *Fas*, *Tnfrsf10a* (TRAIL-R1) and *Tnfr1* (Fig. 6f). Collectively, these findings indicate that SARS-CoV-2 infection promotes post-translational accumulation and activation of caspase-8, driving transcriptional induction of pro-inflammatory signalling in both murine and human lung tissue.

## Caspase-8 drives NF-κB signalling during SARS-CoV-2 infection

cFLIP$_L$/procaspase-8 heterodimers can catalyse the first step of caspase-8 activation, but further processing is precluded, leaving the heterodimeric enzyme complex bound to the DISC. At the DISC, this heterodimer can interact with proximal substrates, inducing NF-κB signalling rather than an apoptosis[38,78–81]. To assess whether NF-κB signalling is essential for driving disease in our model, we infected NF-κB1 knockout (*p50$^{-/-}$*) mice with SARS-CoV-2. Compared to WT controls, *p50$^{-/-}$* animals exhibited significantly reduced viral loads and attenuated weight loss at 3 dpi (Fig. 7a, b). While *p50* deletion disrupts canonical NF-κB signalling, it does not abolish NF-κB pathway activity entirely. Therefore, these findings should be interpreted as reflecting a perturbation, rather than complete loss, of NF-κB signalling, underscoring the importance of intact NF-κB1 function in mediating SARS-CoV-2 induced pathology in this model.

NF-κB signalling downstream of caspase-8 is antagonised by the NEDD4-binding protein 1 (N4BP1), a suppressor of cytokine production that can be cleaved and thereby inactivated by caspase-8[37]. Western blot analysis of infected whole lung tissue confirmed that N4BP1 was cleaved during infection, and this cleavage was dependent on caspase-8 and TNF (Fig. 7c). Quantification of bands confirmed that the proportion of cleaved to full length N4BP1 was increased in SARS-CoV-2 infected WT animals compared to their mock controls. Notably, cleavage of N4BP1 was reduced in SARS-CoV-2 infected *C8/R3$^{-/-}$* or *Tnf$^{-/-}$* mice (Fig. 7d). Collectively, these data reveal that SARS-CoV-2 infection causes an increase in the levels of full-length and p41/p43 caspase-8, which appears to lead to cleavage and thereby inactivation of N4BP1 to trigger an NF-κB-driven pro-inflammatory response. To further investigate the role of TNF in caspase-8 activation, we compared caspase-8 levels between SARS-CoV-2 infected WT and *Tnf$^{-/-}$* animals. Although caspase-8 levels appeared modestly reduced in the absence of TNF (Fig. 7e), quantification and normalization revealed that this difference was not statistically significant (Fig. 7f). This indicates that TNF contributes to, but is not solely responsible for the increase in caspase-8 protein during SARS-CoV-2 infection.

Given the herein proposed non-canonical role of caspase-8 in promoting NF-κB activation thereby enhancing cytokine production, we screened for components of this pathway amongst proteins upregulated in SARS-CoV-2 infected WT mice identified in our proteomics analysis (Fig. 2a–f). This analysis revealed elevated expression of multiple mediators of inflammatory caspase-8 signalling, including caspase-8 itself, RIPK1, RIPK2, RIPK3, ZBP1 and N4BP1 (Fig. 7g). Collectively, these data implicate caspase-8 as a central regulator of the SARS-CoV-2 induced inflammatory response, acting across multiple axes including cytokine production, lipid metabolism, iron homeostasis and complement activation.

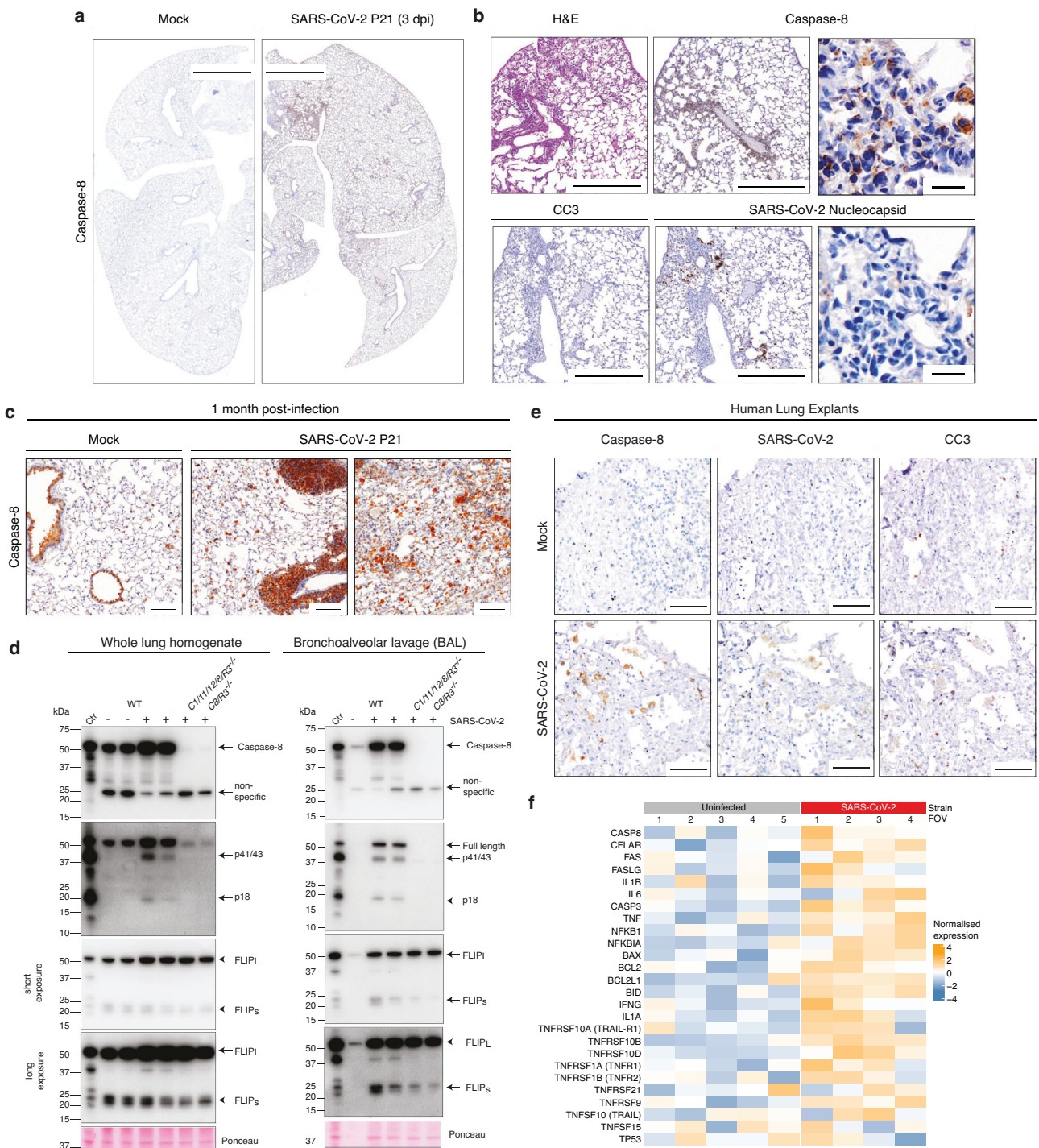

**Fig. 6 | SARS-CoV-2 infection leads to accumulation of caspase-8 in the lungs.**
**a,b** Mice were inoculated intranasally with either mock (media only) or 10⁴ TCID50 of SARS-CoV-2 P21 and lungs were collected at 3 dpi for histological analysis (images are representative of at least 5 mice per group). **a** Representative images of immunohistochemical (IHC) staining for caspase-8 in lungs. Scale bar = 2000 μm. **(b)** Representative images of lungs IHC stained for caspase-8, SARS-CoV-2 nucleocapsid, cleaved (i.e. activated) caspase-3 (CC3) or stained with hematoxylin and eosin (H&E). Scale bar = 500 and 25 μm. **c** Mice were inoculated intranasally with either mock or 10⁴ TCID50 of SARS-CoV-2 P21 and lungs were collected at 1-month post-infection for histological analysis. Scale bar = 100 μm. **d** Western blot

analysis of whole lungs or bronchoalveolar lavage (BAL) from mock, infected WT, *C1/11/12/8/R3⁻/⁻* or *C8/R3⁻/⁻* animals at 3 days after intranasal SARS-CoV-2 P21 infection (10⁴ TCID50). Samples were probed for full-length caspase-8, cleaved caspase-8 and FLIP. Ponceau staining is shown as a loading control. Histology images are representative of at least 3 animals per genotype and condition. **e,f** Human explant tissue obtained through lung biopsy was infected with SARS-CoV-2 or left uninfected (see methods) and **e** analysed histologically through IHC for caspase-8, SARS-CoV-2 nucleocapsid and cleaved-caspase-3. Scale bar = 100 μm (images are representative of at least 3 fields of view), or **f** analysed for gene expression using CosMx™ Spatial Molecular Imaging (Nanostring). Source data are provided as a Source Data file.

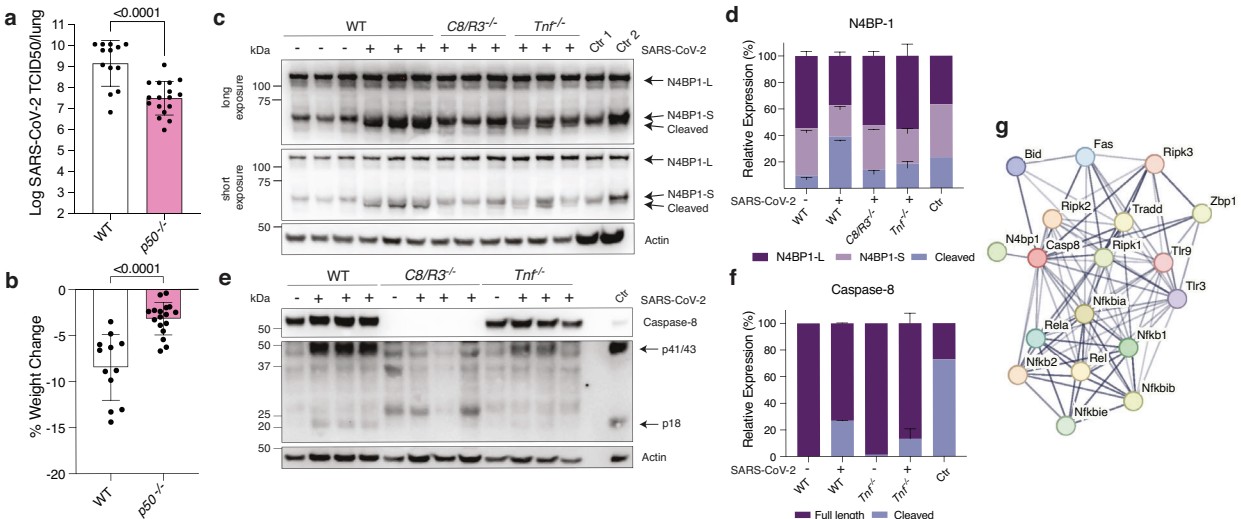

**Fig. 7 | Caspase-8 drives inflammation through NF-κB activation during SARS-CoV-2 infection. a,b** WT and p50 knockout (p50⁻/⁻) mice were infected with 10⁴ TCID50 of SARS-CoV-2 P21 and monitored at 3 days post-infection (dpi) for **a** lung viral burden by TCID50 assay and **b** percent weight change compared to initial weight ((n(WT) = 13; n(p50⁻/⁻) = 17 animals). **c–f** Western blot analysis of whole lungs from mock (media only inoculation) (-) or SARS-CoV-2 P21 infected (10⁴ TCID50) ( + ) WT, *C8/R3⁻/⁻* or *Tnf⁻/⁻* mice at 3 dpi. Each well represents a biological replicate (lung homogenate of one animal). **c** Samples were probed for N4BP1 and actin as loading control. Ctr 1: BMDMs treated with 20 ng of TNF and Ctr 2: BMDMs treated with 200 ng of TNF. **d** Quantification of western blot bands: Intensity of cleaved N4BP1 bands was divided by intensity of full length N4BP1 band and normalised to actin. P values (one-way with multiple comparisons ANOVA) for ratio cleaved/full-length N4BP: Mock vs WT, p = 0.004; WT vs C8/R3⁻/⁻, p = 0.0007; WT vs Tnf⁻/⁻, p = 0.0007; C8/R3⁻/⁻ vs Tnf⁻/⁻, p = 0.9999. **e** Samples were probed for full-length as well as cleaved caspase-8 and actin, the latter used as a protein loading control. **f** Quantification of Western blot bands was performed. Intensity of cleaved caspase-8 bands was divided by intensity of full-length caspase-8 band and normalised to actin. **g** Protein-protein interaction network (STRING) of caspase-8 related proteins with differential enrichment in SARS-CoV-2 infected mice compared to mock controls (from proteomics analysis in Fig. 2). Data are presented as mean ± SD. Unpaired two-tailed Student's t test after log₁₀ transformation (**a**) and unpaired two-tailed Student's t test (**b**) statistical tests were performed; Western blots are representative of 3 independent experiments. Source data are provided as a Source Data file.

## Discussion

We and others have shown that pro-inflammatory cytokines, such as TNF and IL-1β, play critical roles in COVID-19 disease pathogenesis[23,44,45]. While we have previously provided first genetic evidence that canonical inflammasome activation and pyroptosis are not required for IL-1β secretion during SARS-CoV-2 infection in vivo, the molecular mechanisms of this release remained elusive. Our previous work showed that IL-1β release during infection occurred independently of inflammatory caspases-1/−11/−12 and the ASC inflammasome[23]. Here, we found that loss of caspase-8 (alongside removal of RIPK3 to prevent embryonic lethality owing to aberrant necroptosis caused by the absence of caspase-8[53,54,82]) significantly reduced IL-1β levels in the lungs and diminished pathology upon SARS-CoV-2 infection, similar to what was seen in *Il-1β⁻/⁻* mice[23].

Having identified caspase-8's central role in SARS-CoV-2 pathogenesis, it was logical to assume that in addition to its role in promoting IL-1β cytokine levels, it would also contribute to pathology by driving apoptosis. Surprisingly we found that apoptosis is not prominent in the lungs upon infection. Histologically, overt apoptosis was not evident in SARS-CoV-2 infected mice, but western blots revealed incomplete cleavage (i.e. activation) of caspase-3 upon infection. Instead, caspase-7 cleavage was apparent, indicating that caspase-7 may be activated via mitochondrial apoptosis. SARS-CoV-2 infected *C8/R3⁻/⁻* mice showed an increase in both pro-apoptotic BIM and anti-apoptotic BCL-XL compared to infected WT mice, suggesting that compensatory mechanisms regulating apoptosis may be induced. These findings suggest that the significant protection from SARS-CoV-2 lethality observed upon caspase-8 deletion is not due to reduced levels of apoptosis. However, increased levels of cleavage products of caspase-8 were detected in the BAL of SARS-CoV-2 infected mice, and this appeared to partially contribute to the observed increase in CC3.

Whether the apoptotic death of cells in the BAL might impact caspase-8-mediated disease pathology remains unclear.

To determine the extent to which the phenotype of *C8/R3⁻/⁻* animals depends on deletion of the inflammatory driver RIPK3, we characterised disease parameters in *R3⁻/⁻* mice. RIPK3 deficiency alone had no impact on viral loads, weight loss, or cytokine production in infected animals. Spatial transcriptomics analysis of infected lung tissue revealed that some cell cluster proportions were similarly altered in both *C8/R3⁻/⁻* and *R3⁻/⁻* lungs upon infection; however, these changes did not translate into measurable differences in physiological disease severity. To further dissect the contribution of caspase-8 independently of RIPK3, we examined cytokine responses following treatment with the broad-spectrum caspase inhibitor emricasan and the more selective (though not uniquely specific) caspase-8 inhibitor Z-IETD-FMK. Emricasan treatment did not alter viral replication but significantly reduced pulmonary TNF, IL-6, and IL-1β, indicating that caspase-8-dependent cytokine release occurs independently of RIPK3. In contrast, Z-IETD-FMK failed to suppress cytokine production during infection, consistent with recent reports showing variable cytokine modulation by this inhibitor depending on cell type, pathogen, and in vivo pharmacokinetics[83,84]. Notably, emricasan has been shown to effectively inhibit caspase-8 activity within the caspase-8/cFLIP heterodimer[56], a key driver of non-apoptotic, pro-inflammatory signalling, whereas it remains unknown whether Z-IETD-FMK achieves comparable target engagement in this context. Furthermore, recent studies have demonstrated that Z-IETD-FMK can partially inhibit other effector caspases (e.g. caspase-3/−6) and PARP, and that no currently available caspase-8 inhibitor is truly specific[85,86]. The divergent effects observed here likely reflect differences in target selectivity, cell permeability, and tissue distribution, as well as potential off-target activities[87]. Together, these findings underscore the context-

dependent and multifunctional roles of caspase-8 during infection, and suggest that selective inhibition of its non-apoptotic signalling might blunt cytokine production but could also predispose to increased necroptotic cell death. Development and application of truly selective caspase-8 inhibitors or genetic silencing strategies (e.g. siRNA) will be required to formally test this hypothesis.

To elucidate the upstream regulation of caspase-8 in SARS-CoV-2 induced disease, we investigated the roles of TNF and IFN-γ in our model. While previous investigations have implicated both TNF and IFN-γ as primary drivers of SARS-CoV-2 induced pathology, these findings were largely based on in vitro studies or lacked testing of the impact of the inhibition or genetic removal of TNF alone[43]. Here, using knockout mice, we demonstrate that IFN-γ does not play a significant role in SARS-CoV-2 induced pathogenesis, whereas consistent with our prior work, we confirm a central pathogenic role for TNF[45]. In addition to TNFR1, caspase-8 can also be activated by the death receptor FAS, and recent evidence indicates that inhibition of FAS reduces SARS-CoV-2 induced mortality in mice[88]. However, whether the impact of FAS inhibition in this scenario is mediated through a loss of activation of caspase-8 remains to be determined[88].

Given the importance of IL-1β in promoting severe SARS-CoV-2 disease in our model and the mounting widespread evidence for a central role of this cytokine in driving severe COVID-19 in humans[43,66,89–94], we explored the roles of other caspases in addition to caspase-8 that are known to be able to process this cytokine. While we have previously shown that absence of caspases-1, −11 and −12 do not impact disease or viral burdens, here we report that even with the additional loss of RIPK3 (*C1/11/12/R3*[-/-] mice), we did not recapitulate the amelioration of disease observed in mice lacking caspase-8 (and RIPK3). Interestingly, the absence of caspase-1, −11, −12 together with RIPK3 seemed to worsen SARS-CoV-2-induced disease, perhaps through a mechanism that further unleashes the pro-inflammatory function of caspase-8. When caspase-8 was removed alongside with the absence of caspases-1, −11, −12, SARS-CoV-2 induced disease was yet again substantially ameliorated. Notably, we observed a slight diminished protection from SARS-CoV-2-induced pathology in *C1/11/12/8/R3*[-/-] compared to *C8/R3*[-/-] animals. This may reflect a minor role for these three inflammatory caspases in the response of mice to infection with SARS-CoV-2. While caspases-1 and −11 are well known for their roles in the maturation of IL-1β as well as IL-18 and pyroptosis, they have also been implicated in promoting pathogen clearance through modulation of immune cell function and induction of cell-intrinsic defence processes[95,96]. Therefore, it is plausible that the additional deletion of inflammatory caspases in this quintuple knockout impairs certain compensatory mechanisms that would otherwise support protection in *C8/R3*[-/-] mice.

The conundrum that remained was how caspase-8 is driving pathological inflammation in our model of severe SARS-CoV-2 disease, seemingly independent of its ability to induce apoptosis. Notably, we observed that SARS-CoV-2 infection caused an increase in full-length caspase-8 in both hematopoietic and non-hematopoietic cells in the lungs. In line with previous studies in COVID-19 patients[97–99], this increase in caspase-8 also occurred in infected human lung tissue explants in vitro. In our mouse model in vivo, the increase in caspase-8 persisted after the virus was cleared. Western blot analysis provided a more precise dissection of caspase-8 processing. We observed an increase in full-length caspase-8 and caspase-8 p41/43 cleavage products, but only a minor fraction of caspase-8 was found fully processed to its pro-apoptotic p18 form.

Furthermore, the levels of the caspase-8 regulator cFLIP$_L$ were also increased in the lung upon SARS-CoV-2 infection. This observation is in line with previous studies that reported cFLIP$_L$ to be highly expressed in myeloid cells in lungs of COVID-19 patients[100]. Moreover, SARS-CoV-2 infection has also been shown to induce transcriptional changes in genes encoding regulators acting upstream of cFLIP$_L$[91,101].

Taken together with our findings using caspase inhibitors, these results support a central hypothesis that inflammation during SARS-CoV-2 infection is driven by aberrant caspase-8 activity, specifically through the p41/p43 processed form of caspase-8 in complex with cFLIP$_L$. Caspase-8 can activate NF-κB signalling (reviewed in ref. 102) through both cFLIP$_L$ dependent as well as cFLIP$_L$ independent processes[80,103]. Here we show that disruption of NF-κB signalling through p50 deletion significantly reduces viral burden and disease severity, indicating that SARS-CoV-2 induces pathological NF-κB activation. We found that this activation of NF-κB is likely negatively regulated by N4BP1, a protein that can be cleaved by caspase-8 to allow for efficient TNFR1 and TLR-mediated NF-κB transcriptional responses[37].

Our elucidation of the critical role of caspase-8 in SARS-CoV-2 disease pathogenesis raises many questions around the detailed mechanisms governing its elevated expression and ability to promote inflammation in the absence of prominent apoptotic activity. Presumably, increased levels of activated caspase-8 can trigger the release of pathological levels of IL-1β - whether this occurs in the cells that constitute the BAL, which display increased caspase-8 processing that has been associated with the ability of RIPK3 and caspase-8 to cleave pro-IL-1β[50], remains of outstanding interest. Are there effectors encoded by SARS-CoV-2 that directly promote this pro-inflammatory action of caspase-8? Moreover, the accumulation and pro-inflammatory role of caspase-8 seems to be an idiosyncrasy of SARS-CoV-2 infection. Studies with influenza virus suggest this mechanism might be unique. Unlike shown here for SARS-CoV-2, influenza pathogenicity is mainly driven by inflammasome pathways and GSDMD[104], with caspase-8 actually exerting an anti-inflammatory effect[105].

In line with the current health issues emerging from the long-term effects of COVID-19, we observed elevated caspase-8 levels in the lungs of recovered animals at least up to 1-month post-infection. It is tempting to speculate that caspase-8 may continue to drive inflammation long after the virus has been cleared. While long COVID models are now available to us[106], long-term studies on mice deficient for caspase-8 (and RIPK3 or MLKL) may not be possible due to the development of LPR disease. However, cell type specific conditional deletion of Caspase-8 could provide insights into these open questions.

In summary, our current study underlines the central role of caspase-8 in severe SARS-CoV-2 induced pathology. We provide first evidence of the underlying mechanism in vivo and suggest further investigations to untangle the complexities of the regulation of caspase-8 function(s), its potential interplay with viral components in different cellular contexts, and the potential therapeutic implications of manipulating caspase-8's non-apoptotic inflammatory function.

## Methods

### Mice
Male or female WT and gene-targeted mice were bred and maintained in the Specific Pathogen Free (SPF) Physical Containment Level 2 (PC2) Bioresources Facility at The Walter and Eliza Hall Institute of Medical Research (WEHI). All mice except for *Ripk3*[-/-] mice were on a C57BL/6 J background. *Ripk3*[-/-] mice were on a C57BL/6 N background and experiments were performed comparing *Ripk3*[-/-] to WT C57BL/6 N mice (strain information can be found in Supplementary Table 1).

All procedures involving animals and live SARS-CoV-2 strains were conducted in an OGTR-approved Physical Containment Level 3 (PC3) facility at WEHI (Cert-3621) under approval of national regulations. Mice were transferred to the PC3 laboratory for all SARS-CoV-2 infection studies at least 4 days prior to the start of experiments. Mice were age- and sex-matched within experiments (both sexes were used). "Young adult" mice were 6–8 weeks of age at the commencement of experiments, "adults" were 10–12-week-old and "aged" cohorts were 6–8 months old. Experimental mice were housed in individually

ventilated microisolator cages under level 3 biological containment conditions with a 12 h light/dark cycle. All mice were provided standard rodent chow and sterile acidified water *ad libitum*.

### SARS-CoV-2 strains

SARS-CoV-2 VIC2089 clinical isolate (hCoV-19/Australia/VIC2089/2020) was obtained from the Victorian Infectious Disease Reference Laboratory (VIDLR). Viral passages were achieved by serial passage of VIC2089 through successive cohorts of young C57BL/6 J (WT) mice[45]. Briefly, mice were infected with a SARS-CoV-2 clinical isolate intranasally. At 3 dpi, mice were euthanized, and lungs harvested and homogenised in a Bullet Blender (Next Advance Inc) in 1 mL Dulbecco's modified Eagle's medium (DMEM) (Gibco/ThermoFisher) containing steel homogenization beads (Next Advance Inc). Samples were clarified by centrifugation at 10,000 x g for 5 min before intranasal delivery of 30 µL lung homogenate (P2) into a new cohort of naïve C57BL/6 J mice. This process was repeated a further 20 times to obtain the SARS-CoV-2 VIC2089 P21 isolate. Lung homogenates from all passages were stored at −80 °C.

Viral strains used in this study are available upon signing of a Materials Transfer Agreement. Genomic sequences of SARS-CoV-2 passages (P2 and 21) are available at GenBank, accession numbers OP848479-98 (https://www.ncbi.nlm.nih.gov/bioproject/PRJNA995787).

### Infection of mice with SARS-CoV-2 P21

6–8-week-old, 11-week-old or 6–8-month-old mice were anesthetized with methoxyflurane and inoculated intranasally with 30 µL of DMEM containing SARS-CoV-2 P21. Virus stocks were diluted in serum free DMEM to a final concentration of $10^4$ TCID50/mouse. After infection, animals were visually checked and weighed daily for a minimum of 10 days. Mice were euthanized at the indicated times post-infection by $CO_2$ asphyxiation. For histological analysis, animals were euthanized by cervical dislocation. Lungs were collected and stored at −80 °C in serum-free DMEM until further processing.

To ensure robustness of data, results showing viral load and weight loss were pooled from multiple independent infections, each including internal controls. Given the age-dependence of disease severity induced by infection with SARS-CoV-2 in this model, experimental cohorts were prioritized for age-matching even when all genotypes could not be compared in the same experiment. Both sexes were used as we have previously found no sex-dependent differences in clinical, virological or inflammatory endpoints under identical conditions. Accordingly, sex was considered a priori and data were pooled to maximise statistical power and adhere to the 3Rs.

Humane endpoints for survival experiments were defined as follows: a weight loss exceeding 20% of baseline, sustained weight loss of more than 15% for three consecutive days, or a body condition score of 2 or below. Mice were euthanised upon presentation of any of the following clinical signs of illness: ruffled fur, hunched posture, laboured respiration, lethargy, abnormal gait, head tilt, or hind limb paresis.

### Measurement of viral loads by determining 50% tissue culture infectious dose (TCID$_{50}$)

TCID$_{50}$ analysis was performed as previously described[107]. Briefly, Vero African green monkey kidney epithelial cells, purchased from ATCC (clone CCL-81), were seeded in flat bottom 96-well plates ($1.75 \times 10^4$ cells/well) and left to adhere overnight at 37 °C/5% $CO_2$. Cells were washed twice with PBS and transferred to serum-free DMEM containing TPCK trypsin (0.5 µg/mL working concentration). Infected organs were defrosted, homogenized, clarified by centrifugation at 10,000 x g for 5 min at 4 °C and supernatant was added to the first row of cells at a ratio of 1:7, followed by 9 rounds of 1:7 serial dilutions in the other rows. Cells were incubated at 37 °C/5% $CO_2$ for 4 days until virus-

induced cytopathic effect (CPE) was scored. TCID$_{50}$ was calculated using the Spearman & Kärber algorithm as described in[107].

### Histological analysis and immunohistochemical staining

Organs were harvested and fixed in 4% paraformaldehyde (PFA) for 24 h, followed by 70% ethanol dehydration, paraffin embedding and sectioning. Slides were stained with either haematoxylin and eosin (H&E), or immunohistochemically stained with antibodies against CD3 (1:500, Agilent A045201), MPO (1:1000, Agilent A039829), F4/80 (1:1000, WEHI in-house antibody), caspase-8 (clone 1G12; RRID:AB_2490518; 1 g/L produced in-house and used at 1:200 dilution, or clone D35G2; RRID:AB_10545768; Cell Signaling Technology Cat#4790 used at 1:200 dilution; as described in[108]), cleaved caspase-3 (Cell Signaling Technology #9661) or SARS-CoV-2 nucleocapsid (1:4000, abcam ab271180) using the automated Omnis EnVision G2 template (Dako, Glostrup, Denmark). De-waxing of slides was performed with Clearify Clearing Agent (Dako) and antigen retrieval was conducted with EnVision FLEX TRS, High pH (Dako) at 97 °C for 30 min. Primary antibodies were diluted in EnVision Flex Antibody Diluent (Dako) and incubated at 32 °C for 60 min. HRP-labelled secondary antibodies (Invitrogen, Waltham, USA) were applied at 32 °C for 30 min. Slides were counter-stained with Mayer Hematoxylin, dehydrated, cleared, and mounted with MM24 Mounting Medium (Surgipath-Leica, Buffalo Grove, IL, USA). Slides were scanned using an Aperio ScanScope AT slide scanner (Leica Microsystems, Wetzlar, Germany). Where indicated, a histopathological scoring system (0–3) was used by an American board-certified pathologist (Smitha Rose Georgy) who was blinded to the genotypes of the mice to grade histological changes based on H&E staining. The score was based on the average of the following parameters[109]: perivasculitis and marginating inflammatory cells, pleural mesothelial cell hyperplasia, iBALT, alveolar hemorrhage, alveolar septal thickening, proteinaceous debris and fibrin strands in air space, inflammatory cells in the interstitial space/septae and inflammatory cells in the alveolar space. Grades are based on the percentage of lesions: 0 = normal, 1 < 10%, 2 = 10–25%, 3 = 25–50%, 4 > 50.

### Quantification of IHC images

TIF images were exported from Caseviewer for analysis. Analysis was performed by using the custom python pipeline. Briefly, cells were detected using a pre-trained object detection with Star-convex Shapes network (stardist)[110], and scored based on the signal in the DAB channel after colour deconvolution[111]. Images were split into tiles for memory management purposes and processed on a local high-performance computer.

### Lung cytokine and chemokine analysis

Lungs were thawed, homogenized and clarified by centrifugation at 10,000 x g for 5 min at 4 °C. Supernatants were pre-treated for 20 min with 1% Triton-X-100 for viral de-activation and the Cytokine & Chemokine 26-Plex Mouse ProcartaPlex Panel 1 (EPX260-26088-901) was used as described in the manufacturer's instructions. A total of 25 µL of clarified lung samples were diluted with 25 µL universal assay buffer, incubated with magnetic capture beads, washed, incubated with detection antibodies and SA-PE. Cytokines were recorded on a Luminex 200 Analyzer (Luminex) and quantitated by comparison to a standard curve. Analysis of results was performed using R Studio.

### Western blot analysis

Total cell protein was isolated from whole mouse lungs using cell lysis buffer containing 20 mM Tris-HCl, pH 7.5, 135 mM NaCl, 1.5 mM Mg$_2$Cl, 1 mM EGTA, 1% Triton X-100 (Sigma-Aldrich), 10% glycerol, EDTA-free protease inhibitor tablets (Roche, Basel, Switzerland), and phosphatase inhibitor tablets (Roche). Absolute protein content of clarified lysates was determined using the BCA Protein Assay (Pierce), and equal

quantities (10–30 μg) of total protein were separated under denaturing and reducing conditions (with 5% β-mercaptoethanol) using 4–12% SDS-PAGE gels (Life Technologies). Proteins were transferred onto nitrocellulose or PVDF membranes using Mini Trans-Blot Cell (Biorad), blocked with 5% skim milk (Devondale, Brunswick, Australia) in TBS with 0.05% Tween-20 (TBST) for 30 min. Proteins were detected using the following primary antibodies: rabbit anti-caspase-3 (9662, Cell Signaling Technology), rabbit anti-cFLIP (1:1000, D5J1E, Cell Signaling Technology), rat anti-caspase-8 (3B10, in-house), rabbit anti-cleaved (i.e. activated) caspase-8 (D5B2, Cell Signaling Technology) and rabbit anti-N4BP1 (Abcam ab133610). HRP-conjugated goat secondary antibodies (anti-rabbit IgG, anti-rat IgG) (Southern Biotech, Birmingham, AL, USA) were then applied to membranes, which were subsequently incubated with Immobilon Western Chemiluminescent HRP Substrate (Merck Millipore) and imaged using a ChemiDoc Touch Imaging System (Bio-Rad).

A positive control for caspase-8 and caspase-3 activation was prepared by treating mouse bone marrow derived macrophages (BMDMs) with 50 ng/mL of IFN-γ overnight, followed by 21 h incubation with 50 ng/mL lipopolysaccharide (LPS). Positive controls for N4BP1 consisted of BMDMs treated with either 20 ng or 200 ng of TNF.

### Emricasan and Z-IETD treatment of mice
Mice were treated with either vehicle (10% DMSO, 40% PEG300, 5% Tween-80 in saline) or emricasan (60 mg/kg body weight) intraperitoneally twice daily for 3 days, starting 1 h prior to infection with SARS-CoV-2 P21.

For Z-IETD studies, animals were administered 6 mg/kg body weight Z-IETD in 15% DMSO/PBS or vehicle daily for 3 days starting 1 h prior to infection with SARS-CoV-2 P21.

### RNA sequencing
Mice were infected with SARS-CoV-2 P21and the left lung lobe was collected for TCID50 assay, while the right lobe was stored in RNA-later™ Stabilization Solution (CAT# AM7020; Thermo Fisher Scientific, Waltham, MA, USA). Lungs were homogenized using pellet pestles and RNA extraction was performed using the Isolate II RNA mini kit according to the manufacturer's instructions (Cat# BIO-52072; Meridian Bioscience, Cincinnati, OH, USA). An input of 100 ng of total RNA was prepared and indexed for Illumina sequencing using the TruSeq RNA sample Prep Kit (Cat# RS-122-2001; Illumina, San Diego, CA, USA) with RiboGlobin depletion according to the manufacturer's instructions. Each library was quantified using the Agilent Tapestation (using RNA ScreenTape [Cat# 5067-5576] on a 2200 TapeStation system (Cat# G2964AA; Agilent Technologies, Waldbrunn, Germany)). The indexed libraries were pooled and diluted to 1.5 pM for paired end sequencing (2 × 81 cycles) on a NextSeq 500 instrument using the v2 150 cycle High Output kit (illumina) according to the manufacturer's instructions. The base calling and quality scoring were determined using Real-Time Analysis on board software v2.4.6, while the fastq file generation and de-multiplexing utilized bcl2fastq conversion software v2.15.0.4.

### Bioinformatic analysis of RNAseq data
Quality control analysis of RNA-seq fastq files was performed using FastQC (v0.11.9, https://www.bioinformatics.babraham.ac.uk/projects/fastqc/). Individual FastQC files were consolidated into a single report using MultiQC (v1.12, https://multiqc.info/). A genome index was built using the Mus musculus GRCm39.dna.primary_assembly fasta file (http://ftp.ensembl.org/pub/release-106/fasta/mus_musculus/) using the Rsubread[112] (v4.2) buildindex function. Reads were aligned to the reference mouse genome using the Rsubread align function and gene counts were quantified from the resultant BAM files using Subread featureCounts[113] (v2.0.3) and the Mus musculus GRCm39.106 reference GTF file (http://ftp.ensembl.org/pub/release-106/gtf/mus_musculus/). The gene count matrix was analyzed in RStudio[114] (v1.4.1743-4, R 4.2.0) and samples were split into one of six experimental groups:

1. WT mock (n = 4)
2. *C1/11/12/R3*⁻ᐟ⁻ mock (n = 4)
3. *C1/11/12/8/R3*⁻ᐟ⁻ mock (n = 4)
4. WT infected with 10⁴ TCID50 SARS-CoV-2 P21 (n = 6)
5. *C1/11/12/R3*⁻ᐟ⁻ infected with 10⁴ TCID50 SARS-CoV-2 P21 (n = 6)
6. *C1/11/12/8/R3*⁻ᐟ⁻ infected with 10⁴ TCID50 SARS-CoV-2 P21 (n = 6)

The count matrix was filtered using the edgeR[115] (v3.15) filter-ByExpr function and including the grouping variable to apply filtration relative to the smallest group size (n = 4). Gene expression distributions were normalized using the trimmed mean of M-values (TMM) method implemented by the edgeR calcNormFactors function. Sample clustering was assessed by multidimensional scaling (MDS) analysis. Heteroscedasticity and sample-level variation was removed from the data by applying the limma[116] (v3.15) voomWithQualityWeights function. Linear models were fitted for comparisons between infected WT infected vs. infected *C1/11/12/8/R3*⁻ᐟ⁻ mice and infected *C1/11/12/R3*⁻ᐟ⁻ mice vs. infected *C1/11/12/8/R3*⁻ᐟ⁻ mice using the limma lmfit and contrasts.fit functions.

Following empirical Bayes moderation, significance testing was performed using the limma treat function, with significantly DE genes defined as those with an FDR-adjusted p-values below 0.05 and an absolute log2 fold-change of greater than 1. The Broad Institute's Mouse Hallmark Gene Sets collection of signatures (https://bioinf.wehi.edu.au/software/MSigDB/) was used to perform gene set testing with the limma camera method.

### Proteomics
Lungs of mock controls and SARS-CoV-2 P21 infected animals were washed three times with ice-cold TBS, lysed in 2% sodium deoxycholate (SDC) (v/v) and 100 mM Tris-HCl (pH 8.5), and boiled immediately. After sonication, protein amounts were adjusted to 20 μg using the BCA protein assay kit. Samples were reduced with 10 mM (TCEP), alkylated with 40 mM 2-chloroacetamide, and digested with trypsin and lysC (1:50, enzyme/protein, w/w) overnight. Peptides were desalted using SDB-RPS-stage tips. Peptides were resolubilised in 5 μL 2% (v/v) acetonitrile (ACN) and 0.3% (v/v) trifluoroacetic acid (TFA) and 200 ng were injected into the mass spectrometer.

### LC-MS
Samples were loaded onto a C18 fused silica column (inner diameter 75 μm, OD 360 μm × 15 cm length, 1.6 μm C18 beads) packed into an emitter tip (IonOpticks) using pressure-controlled loading with a maximum pressure of 1,500 bar with the Neo Vanquish liquid chromatography system (Thermo Fisher Scientific) coupled to the MS (Orbitrap Astral, Thermo Fisher Scientific). Peptides were introduced onto the column with buffer A (0.1% FA) and 4% buffer B (80% ACN, 0.1% FA) followed by an increase of buffer B to 34% for 20 min, and then to 100% for 3 min at a flow rate of 400 nL/min.

A data-independent acquisition MS method was used in which one full scan (380–980 m/z, R = 240,000) at a target of $5 \times 10^6$ ions was first performed, followed by 300 windows with a resolution of 80,000 (at m/z 524) where precursor ions were fragmented with higher-energy collisional dissociation (collision energy 25%) and analysed with an AGC target of $8 \times 10^4$ ions and a maximum injection time of 3 ms in profile mode using positive polarity.

Data were analysed using R-Studio and interactions between proteins within lists of interest were examined using STRING[117].

### Gene ontology analysis
To identify biological processes enriched among differentially expressed genes, we performed Gene Ontology (GO) enrichment

analysis separately for upregulated and downregulated genes using the clusterProfiler R package (v4.8.1). Genes were classified as significantly upregulated or downregulated based on the following cutoffs: $|\log_2$ fold change$| > 1$ and $-\log_{10}$(adjusted p-value) $> 1.3$. Upregulated and downregulated gene sets were independently converted to Entrez IDs using bitr() (OrgDb: org.Mm.eg.db), and enrichment for GO Biological Process terms was conducted with enrich GO (Benjamini–Hochberg adjusted p-value < 0.1, q-value < 0.2).

## MERFISH imaging
MERFISH spatial transcriptomics (Vizgen) was performed according to the manufacturer's protocol (Vizgen, 9160112 Rev D). Briefly, formalin-fixed paraffin-embedded mouse lung tissues were placed onto MER-SCOPE coverslips (Vizgen, 10500102), air-dried for 2 h at room temperature, incubated at 55 °C for 15 min before storing at -20 °C until further processing. After deparaffinisation, decrosslinking, cell boundary staining and RNA anchoring, tissues were embedded in a thin polyacrylamide gel and subjected to proteinase K-containing clearing solution until they became transparent. The tissues underwent autofluorescence quenching using MERSCOPE Photobleacher (Vizgen, 10100003) for 4 h at room temperature, followed by hybridisation with the custom-designed 303 gene probe set (Vizgen, 20300008) at 37 °C for 48 h. MERFISH measurements were conducted on the MERSCOPE platform as described in the MERSCOPE User Guide (Vizgen, 91600001 Rev H). For processing MERFISH data, RNA molecules were decoded using MERlin (Vizgen, software version v233).

## Cell segmentation and MERFISH data processing
Custom cell segmentation was performed with Cellpose using a human-in-the loop approach[118,119]. The images corresponding to the 3 cell boundary stains at the fourth imaging plane (z = 3) was selected for cell segmentation. All the channels were combined into a single image using a maximum intensity projection "Cellbound_MAX". The "Cellbound_MAX" in combination with DAPI was used for training a cell segmentation model. The Vizgen Post-Processing tool (VPT) was used for extracting 53 image patches, which were split into 48 for manual annotation and 5 for test dataset. VPT applied contrast-enhancement using CLAHE normalisation with a clip limit of 0.01 to all images before training and inference. The pretrained cyto2 models were used for training a custom model using the following settings: Epochs: 100, Learning rate: 0.1, Weight decay: $1 \times 10^{-4}$. The trained model had an average precision score of 0.81. VPT was used to apply cell segmentation using the cellpose2 plugin (vpt-plugin-cellpose2) using the following settings: Cell Diameter: 93.11 px, Flow Threshold: 0.6, Cell Probability Threshold: -4, Minimum mask size: 100 px. VPT was used to regenerate the cell-by-gene matrix, following custom cell segmentation, which was used for downstream analysis.

## Detection of SARS-CoV-2 infected areas
QuPath[120] (v0.4.3) was used for detecting areas with SARS-CoV-2 infection. The image channel corresponding to SARS-CoV-2 nucleocapsid phosphoprotein was imported into QuPath, and a pixel classifier was trained to detect the CoV-2 infected areas. The classifier was applied on all images, and the resulting annotations were exported as geojson files. The annotations were imported as shapely[121] (v2.1) polygon geometry in Python (3.10) and subsequently converted into MERSCOPE micron coordinate space. The distances between cells and CoV-2 infected area were computed using geopandas[122] (v0.14.4).

## MERFISH data quality control and analysis
After performing cell segmentation, we filtered out all cells with total transcript count less than 12, number of unique gene transcripts less than 10, or percentage of counts from blank probes greater than 5%, resulting in a total of 182,757 cells and 300 genes (omitting 3 genes used for viral protein detection) for analysis. Each of the four samples were normalized

using Seurat[123] (v5.3.0) SCTransform(), and subsequently integrated with IntegrateLayers(method = HarmonyIntegration)[124] (Harmony v1.2.3). Clustering of the integrated data was performed by the Louvain algorithm using FindNeighbors() and FindClusters(resolution = 0.2), resulting in 15 clusters. Clusters were manually annotated using cluster marker genes (obtained via FindAllMarkers()) by reference to scRNA-seq data from the LungMAP Mouse Lung CellRef atlas[125] (https://www.lungmap.net/dataset/?experiment_id=LMEX0000004397). We identified 11 clusters corresponding to reference cell types (five immune cell clusters: 'myeloid cells', 'T-cells/natural killer cells', 'T-cells/interstitial macrophages', 'B-cells', and 'mature dendritic cells'; one endothelial cell cluster: 'endothelial cells'; three epithelial cell clusters: 'epithelial alveolar cells', 'ciliated/deuterosomal cells', and 'ciliated/deuterosomal + secretory cells'; and two mesenchymal cell clusters: 'pericytes' and 'mesenchymal cells'), and four clusters that did not correspond to reference cell types, which we labelled by their top differentially expressed gene ($Fosb^+$, $Ptgs2^+$, $Ptx3^+$ and $Ifnb1^+$ cells) (Fig. S2c–e).

To investigate which cell types the $Fosb^+$, $Ptgs2^+$, $Ptx3^+$ and $Ifnb1^+$ clusters corresponded to, we repeated the above-described normalization, integration and clustering with transcriptomic counts for the $Fosb$, $Ptgs2$, $Ptx3$ and $Ifnb1$ genes omitted, i.e. using 296 genes. Using the same clustering parameter resolution=0.2, we recovered the 11 known cell type clusters above, but found the $Fosb^+$, $Ptgs2^+$, $Ptx3^+$ and $Ifnb1^+$ clusters were no longer present, and observed that expression of these four genes was not specific to any specific known cell type (Fig. S2d). Subsequent analysis used the 11 cell clusters based on 296 genes.

We identified 10 spatial niches, each defined by a different composition of spatially adjacent cell types, using Seurat BuildNicheAssay() based on the 30 nearest-neighbouring cells. For both cell type clusters and spatial niches, we investigated the relationship with SARS-CoV-2 infection by computing the relative proportions of cell types and niches, stratified by the distance of each cell to the nearest CoV-2 infected area. We investigated whether there was elevated expression of $Fosb$, $Ptgs2$, $Ptx3$ and $Ifnb1$ genes in areas of SARS-CoV-2 infection by categorising cells as having high expression if normalized (via SCTransform) gene expression per cell was greater than a threshold of 5.

## Human lung tissue procurement
Written informed consent was obtained from all patients by the Victorian Cancer Biobank prior to inclusion in the study, according to protocols approved by the WEHI Human Research Ethics Committee (HREC, 10/04LR). Donor 1 (CosMx) was a 63-year-old female never smoker who contributed healthy lung tissue, collected after lobectomy of the right lower lobe. Donor 2 (Histology) was a 58-year-old female current smoker who contributed healthy lung tissue, collected during lobectomy of the left lower lobe.

## Human lung tissue explant culture
Human lung tissue was stored on ice in DMEM/F12 media (Gibco) supplemented with 1 mg/mL of penicillin and streptomycin (Invitrogen) until processing. Samples were washed in Dulbecco's phosphate-buffered saline (DPBS) with 1% (v/v) antibiotic/antimycotic and processed immediately. Tissue was cut into 2-4 mm³ explants while being kept immersed in chilled antibiotic DPBS. Explants were positioned on pre-soaked gelatin sponges (Johnson & Johnson, Gelfoam) placed in 24-well plates, one sponge per well, and overlaid with 500 μL of explant medium. Explant medium consisted of DMEM/F-12 + GlutaMAX (Gibco) supplemented with ITS (Gibco #41400), B27 (Gibco #17504-044), 0.2% Heparin (Sigma #H4874), 0.1 μg/ml EGF (Sigma #E9644), 25 ng/ml hHGF (R&D #294-HGN), 50 ng/ml hFGF-10 (R&D #345-FG), 100 IU/ml IL-2 (R&D #10453-IL), 100 nM A83-01 (Tocris) and Penicillin/streptomycin (Gibco). Medium was filter-sterilised (0.22 μm) and pre-warmed before use. Complete medium was replaced daily; on day 1 an

additional 500 μL of medium was gently added to each well without submerging the sponge. Plates were incubated at 37 °C, 5% CO$_2$, with medium replaced daily.

## Cellularity estimate

For multiplicity-of-infection (MOI) calculations, representative explants were minced then digested for 45 min at 37 °C with 2 mg/mL collagenase I (Worthington, #LS004197) and 200 U/mL deoxyribonuclease (Worthington, #LS002140) in 0.2% D-glucose (Sigma) in DPBS (Gibco), according to previously published protocols[126]. The cell suspension was filtered through a 100 μm cell strainer and washed with 2% FCS-PBS, followed by red blood cell lysis and further washing with 2% FCS-PBS to obtain a single cell suspension and counted using a haemocytometer. Cell yields averaged $2 \times 10^5$ cells per explant and this value was subsequently used for MOI calculations for every experiment.

## Infection protocol

On experimental day 3, explants were transferred to a PC3 facility at the Peter Doherty Institute and inoculated with 500 μl of hCoV-19/ Australia/VIC17991/2020 (GISAID: EPI_ISL_779606) diluted in culture medium at an MOI of 2.5 ($2 \times 10^5$ cells well$^{-1}$). Infection proceeded for 4 h at 37 °C, 5% CO$_2$. Explants were then washed five times with warm DPBS and returned to fresh medium. Thereafter, 50% medium changes (250 μL) were performed daily. At 5 dpi, whole sponges were fixed in 10% neutral-buffered formalin for 24 h within the PC3 facility, transferred to 80% ethanol overnight and processed for paraffin embedding.

## Downstream analyses of cultured human lung tissue explants

Formalin-fixed paraffin-embedded (FFPE) sections were subjected to immunohistochemistry (IHC) as described above. Consecutive sections were processed for CosMx™ Spatial Molecular Imaging (Nanostring) using the 1000-plex RNA panel according to the manufacturer's instructions. Imaging was performed on the CosMx instrument with the Technology Access Program (Nanostrin, Seattle).

CosMx spatial transcriptomics data were analysed by comparing uninfected and SARS-CoV-2 infected conditions. These data are publicly available through Zenodo (https://doi.org/10.5281/zenodo. 15597679). Analysis was focused on 42 selected genes, identified through previous mouse studies. Uninfected and infected samples were separated in five and four different fields-of-view (FOVs), respectively. To address variability in total gene detection across FOVs, we normalized the gene expression data for each gene within individual FOVs. Specifically, we first calculated total gene detection by summing all the detected transcripts within each FOV in uninfected and alpha strains. The detection counts for each gene were then normalized by dividing by the corresponding total gene detection of that FOV. The resulting proportions were scaled by multiplying by a factor of one million and then log2-transformed (Eq. 1):

$$\text{Normalised detection per gene} = \log_2 \left( \frac{\text{Gene detections in FOV}}{\text{Total detections in FOV}} \times 10^6 \right) \quad (1)$$

To illustrate the difference between the uninfected and SARS-CoV-2 infected samples, we visualized the normalized expression values of the 42 genes in each relevant FOV using the ComplexHeatmap package v2.24.0 in R (version 4.5.0).

## Quantification of data and statistical analysis

Statistical analyses were performed using Prism v9.3.1 software (GraphPad Software, Inc.). Unpaired two-tailed t-tests were used for normally distributed data for comparisons between two independent groups. Data that violated the assumption of normality were transformed by generating log$_{10}$ prior to statistical analysis. Bars in figures represent either the mean ($\pm$ SD) or median of normally or non-normally distributed datasets, respectively, and each symbol represents one mouse. Sample sizes (n), replicate numbers and significance can be found in the figures and figure legends. For all statistical significance indications: *$p < 0.05$; **$p < 0.01$; ***$p < 0.001$; ****$p < 0.0001$.

Statistical analysis of cytokine data consists of two-sided Wilcoxon rank sum test between group medians, with Bonferroni adjustment for multiple comparisons. Boxplots in figures depict the median and interquartile ranges. Loess smoothing was applied to the infection time course data, with the shaded area indicating 95% confidence intervals. For the RNA-seq data, statistical significance was defined based on an FDR-adjusted p-value < 0.05 and an absolute log$_2$ fold-change > 1.

## Ethics Statement

Animal experiments were performed at The Walter and Eliza Hall Institute of Medical Research (WEHI). Procedures and mouse strains were reviewed and approved by the WEHI Animal Ethics Committee (ethics number 2020.016 and 2024.006). All animal experiments were conducted in accordance with the Prevention of Cruelty to Animals Act (1986) and the Australian National Health and Medical Research Council Code of Practice for the Care and Use of Animals for Scientific Purposes (1997).

## Reporting summary

Further information on research design is available in the Nature Portfolio Reporting Summary linked to this article.

# Data availability

All data is available within this paper or as Supplementary Information. Source data are provided with this paper. The mass spectrometry proteomics data have been deposited to the ProteomeXchange Consortium via the PRIDE[127] partner repository with the dataset identifier PXD057656 (https://www.ebi.ac.uk/pride/archive/projects/ PXD057656). CosMx data from human patient explants is available in Zenodo (https://doi. org/10.5281/zenodo.15597679). https://zenodo.org/records/15597680. MERFISH data and processed objects used for performing the analysis shown in the figures in the manuscript are available from Zenodo (https://doi.org/10.5281/zenodo.15719666). The RNA-seq data generated in this study have been deposited at GEO (GSE282408) and are publicly available as of the date of publication. Source data are provided with this paper.

# Code availability

Codes for Merscope processing and Cellpose cell-segmentation are available from Zenodo (https://doi.org/10.5281/zenodo.15719804).

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

## Acknowledgements

We thank Drs Seth Masters, Christopher J. Tonkin and Johanna Groom for knockout mouse strains as well as Natalia Mora Torres, Janzelle Marasigan and Thomas Kapitelli for excellent animal husbandry support. The authors acknowledge the resources, scientific contribution and technical expertise of the Advanced Histotechnology Facility at Walter and Eliza Hall Medical Institute. We thank Ruvimbo Mishi and the WEHI Spatial Omics Platform for their assistance. The authors acknowledge the contribution and assistance of Melbourne Health through its Victorian Infectious Diseases Reference Laboratory at the Doherty Institute, in providing our laboratory with isolated SARS-CoV-2 material. This work has been supported by: Australian National Health and Medical Research Council Australia Investigator grant GNT1175011 (M.P.), Bellberry-Viertel Senior Medical Research Fellowship (M.Doe.). Medical Research Future Fund (MRFF) Post-Acute Sequelae of COVID-19 grant (APMRF2032843 to M.Doe., Australian National Health and Medical Research Council (NHMRC) ideas grant (1183070 to J.E.V.) and NHMRC investigator grants (2008692 to J.E.V) and (1020363 to A.S.), Victorian Government Department of Health and Human Services 'Victorian COVID-19 Research Fund' (M-L.A-L., M.P.) and the MAWA trust - developing alternatives to animal research (M-L.A-L., M.P.).

## Author contributions

S.M.B. designed and performed experiments, analysed data, generated figures and coordinated all work. R.Ban, J.P.C., J.S., L.Wa. and K.B. designed and performed experiments. D.S. and D.V.L.R. analysed RNA-seq data and generated figures. L.M., M.Da. and L.S. and performed experiments. S.W. oversaw the RNA sequencing and supported analysis. M.C.T and K.S. oversaw and performed proteomics analysis. D.B., M.J.G. and M.L.A.L. performed or oversaw lung explant experiments. X.J and B.P analysed CosMx spatial data. A.J.M. analysed mouse MERSCOPE data. R.K.H.Y. designed MERSCOPE gene panel, generated data and optimised experiments. I.Z. and P.R. performed custom cell segmentation for MERSCOPE image analysis. R.Bow. initiated the mouse lung model MERSCOPE resource and helped bioinformatics analysis. L.Whi. provided the code to analyse IHC data. A.L.S. provided reagents and protocols for IHC. S.R.G. and A.M. scored histological images. J.E.V., J.P. and S.C. optimized, performed, and interpreted western blot data. K.C.D. and C.C.A. planned and oversaw experiments. M.J.H. and A.S. contributed critical ideas and gene-targeted mice and were involved in analysing data. M.P. and M.Doe. supervised all the work. S.M.B., A.S., M.P. and M.Doe. wrote the manuscript with input from other authors.

## Competing interests

A.S. is a consultant for Dimericon which are generating compounds to modulate the functions of caspase-8 and FLIP. A.L.S. has contributed to the development of necroptosis pathway inhibitors in collaboration with Anaxis Pharma Pty Ltd. The other authors declare no competing interests relating to this work.

## Additional information

Stefanie M. Bader ®[1,2] ✉, Lena Scherer[1,2], Reet Bhandari ®[1,2], Allan J. Motyer[1,2], James P. Cooney[1,2], Liana Mackiewicz ®[1], Merle Dayton[1], Dylan Sheerin[1,2], David V. L. Romero[1,2], Jan Schaefer ®[1,2], Jiyi Pang[1,2], Siqi Chen[1,3], Kael Schoffer[1], Le Wang[1,2], Xinyi Jin[1,2], Daniel Batey[1], Raymond K. H. Yip ®[1,2], Ishrat Zaman[1], Pradeep Rajasekhar ®[1,2], Matthew J. Gartner[4], Stephen Wilcox[1,2], Lachlan Whitehead ®[1,2], Smitha Rose Georgy ®[5], Ana Maluenda[1], Kathryn C. Davidson[1,2], Cody C. Allison[1], Rory Bowden ®[1,2], Kerstin Brinkmann ®[1,2], Marie-Liesse Asselin-Labat ®[1,2], Belinda Phipson ®[1,2], Maria C. Tanzer ®[1,2], Marco J. Herold ®[1,2,6,7], Andre L. Samson ®[1,2], James E. Vince ®[1,2], Andreas Strasser ®[1,2], Marc Pellegrini[1,2,8] ✉ & Marcel Doerflinger ®[1,2] ✉

[1]The Walter and Eliza Hall Institute of Medical Research, Melbourne, Victoria, VIC, Australia. [2]Department of Medical Biology, University of Melbourne, Melbourne, VIC, Australia. [3]College of Life Sciences, Nankai University, Tianjin, China. [4]Department of Microbiology and Immunology, the University of Melbourne at the Peter Doherty Institute for Infection and Immunity, Melbourne, VIC, Australia. [5]Section of Anatomic Pathology, Melbourne Veterinary School, Faculty of Science, University of Melbourne, Werribee, Victoria, VIC, Australia. [6]Olivia Newton-John Cancer Research Institute, Heidelberg, Victoria, VIC, Australia. [7]School of Cancer Medicine, La Trobe University, Heidelberg, Victoria, VIC, Australia. [8]Centenary Institute and University of Technology Sydney, Faculty of Science, School of Life Sciences, Sydney, NSW, Australia. ✉e-mail: bader.s@wehi.edu.au; m.pellegrini@centenary.org.au; doerflinger.m@wehi.edu.au

