## [Transparent Peer Review file · Nature Communications]

Non-apoptotic caspase-8 is critical for orchestrating exaggerated inflammation during severe SARS-CoV-2 infection.

Corresponding Author: Dr Marcel Doerflinger

Version 0:

Reviewer comments:

Reviewer #1

(Remarks to the Author)

Programmed cell death (PCD) and caspase activation has been shown to contribute to severe disease after coronavirus infection. Pro-inflammatory responses induced after SARS-CoV-2 infection contribute to PANoptosis in myeloid cells, necroptosis in infected epithelial cells and sequential activation of PANoptosis in uninfected bystander epithelial cells. The detrimental effects of pro-inflammatory responses after SARS-CoV-2 infection has been partially attributed to PCD mediated by caspase activation. Caspases have been reported to contribute to severe disease mostly by mediating PCD. Caspase 4/11 was reported to promote severe disease independent of PCD. Similarly, Bader et al. showed that caspase 8 contributes to heightened inflammatory response after SARS-CoV-2 infection. The authors demonstrated by genetic knockout (KO) mice, that caspase 8 promotes severe disease by cleaving negative regulator of NF- κ B N4BP1. While most of the experiments are technically sound, I have a few major concerns.

1. While caspase 8 KO mice are embryonic lethal, the use of C8/R3^{-/-} mice may confound the author's conclusion a RIPK3 activates necroptosis as noted by the authors, where necroptosis was reported to contribute to severe disease in infected animals (PMID: 38996010). The phenotype observed by the authors using R3^{-/-} mice does not agree with this previous study. It is possible that RIPK1 may compensate the role of RIPK3. The authors should characterize RIPK1-mediated necroptosis in R3^{-/-} mice to rule out that the phenotype observed is due to reduced necroptosis in R3^{-/-} mice.
2. The authors partially addressed point 1 by using caspase inhibitor. However, a broad spectrum caspase inhibitor was used. The phenotypic observed maybe contributed by the inhibition of other caspases. For example, inhibition of caspase 6 has been shown to protect against SARS-CoV-2 infection (PMID: 35922005). To rule out the contribution of other caspases, the authors should use caspase 8 specific inhibitor such as Z-IETD-FMK.
3. The section heading in line 227 is logically flawed. You cannot say that C8 is not driving apoptosis (CC3 still present in Fig. 3D) and at the same time saying C8-driven apoptosis does not significantly contribute to disease. It should be changed to "apoptotic-independent C8 functions contribute to...." or something similar.
4. I do not agree with the author's interpretation that the level of CC3 is the same in Fig. 3D between WT and C3/R8^{-/-} mice. I am not sure this difference would translate to any phenotypic difference but there is a clear visual difference in the band intensity. Uninfected C8/R3^{-/-} mice should also be shown. Also, the authors should show that the apoptosis observed in C8/R3^{-/-} mice are mediated by intrinsic apoptosis pathway to support their claim.
5. Why did the authors choose to use C1/11/12/8/R3^{-/-} mice instead of C8/R3^{-/-} mice for the experiments shown in Fig. 4I-J and Fig. 5?
6. To show that the detrimental effects of caspase 8 is mediated through N4BP1 cleavage, the authors should show that C8/R3/N4BP1^{-/-} mice are no longer protected.

Reviewer #2

(Remarks to the Author)

Bader et al use a SARS-CoV2 mouse model of infection to examine the non-apoptotic role of caspase-8 in the pathological inflammatory response. The topic is impactful and interesting. There are some conclusions in the text that need to be tempered given the experimental evidence used to support them. Most importantly though, some of the genetic comparisons of infected mice throughout the manuscript have major problems that weaken the authors central claims.

Major points:

- In Figure 4, the authors compare WT mice to C1/11/12/8/R3^{-/-} mice. This comparison is simply uninterpretable. I understand that BSL3 conditions limit the number of mice and cohorts that can be studied in 1 infection experiment, but that doesn't change the fact that comparing a quintuply KO'ed mouse to a WT is simply not appropriate. The individual KO's of these mice may have competing effects, or a subset of the KO's may have synergistic or competing effects - the possibilities that convolute interpretation are immense. Moreover, I don't feel this experiment adds more to the conclusion the authors are already attempting to make earlier – that Casp8 has a major effect on the SARS-CoV2 inflammatory response.
- Statement in lines 167-169 should be tempered. It's highly unlikely that CD3+B220⁺ DN T cells haven't developed at all in Ripk3/Casp8-DKO mice beyond 11 weeks. It might be that these cells haven't accumulated yet to the point that gross lymphadenopathy is readily observable, but the cells are undoubtedly there. As such, the authors should modify this statement to take into account that there could still be effects from these cells on the mice.
- In Fig 1A/B vs 1C/D – why were WT, Ripk3-KO, and Ripk3/Casp8-DKO not infected in parallel? This is a bit concerning because the WT mice have TCID50 values differing by 10-100x in 1A vs 1C and similarly the average weight loss for WT in 1B is ~7% whereas it's >10% for WT in 1D. This makes it difficult to exclude the lack of an effect for Ripk3-KO alone as compared to a significant effect for Ripk3/Casp8-DKO.
- In addition, main Figures 1A-D and G: C8/R3^{-/-} vs WT (1A-B) and R3^{-/-} vs WT (1C-D) are distinct experiments, yet ELISA data in 1G appears to be a single experiment. If an experiment was performed with all 3 genotypes, thus using the same WT mice, for 1G, this data would be more effective in place of 1A-D. There is an analogous issue in Main Fig. 4A-D and E.
- Statistical significance in 1G?
- In 1K, emricasan treatment of mice – why don't the mice fare worse with emricasan treatment since there's clearly bountiful TNF and the addition of emricasan would be predicted to trigger necroptosis? Also, the authors do not prove that emricasan's effect is via casp8, thus line 203 conclusion should not specifically make a conclusion about caspase-8 versus other caspases.
- Lines 205-206 make a concluding sentence that Casp8 drives severe disease dependent on TNF, but the underlying data only show that Casp8 or TNF loss separately affect COVID19 disease and do not necessarily support a single linear pathway by which they influence disease. As such, this conclusion should be modified.
- Fig 4 is not clearly interpretable given the lack of single KO controls
- P50 deletion does not 'abrogate' nfkb signaling and may lead to complex effects including gain-of-function dimerization events among other canonical nfkb TFs like P65 or cRel. These experiments are thus not conclusive as to nfkb signaling.
- Figures 5A and B: Labeling on top of graphs is confusing. 5A is labeled WT vs C1/11/12/8/R3^{-/-} with WT functioning as the numerator in the fold change calculation. However, 5B is labeled C1/11/12/8/R3^{-/-} vs. C1/11/12/8/R3^{-/-} with C1/11/12/8/R3^{-/-} functioning as the numerator in the fold change calculation. This should be corrected to ensure clearness. Furthermore, given that the focus is on differences due to CASP8 presence, I would recommend that analyses compare C1/11/12/8/R3^{-/-} vs WT or C1/11/12/8/R3^{-/-} (thus C1/11/12/8/R3^{-/-} would be the numerator for fold change calculations) to demonstrate how CASP8 loss upregulates or downregulates responses to SARS-CoV-2 infection.
- Figure 5C: Label on top of figure should be reversed to read 'Infected vs. Mock' as the figure describes proteins upregulated and downregulated in infection compared to mock.

Reviewer #3

(Remarks to the Author)

In this study, Bader SM et al investigated the role of caspase-8 in inflammation and pathogenesis upon SARS-CoV-2 infection. The authors show that animal survival was improved and the expression inflammatory mediators were attenuated in the caspase-8 and RIPK3 double KO mice when compared to the WT mice. Next, they compared SARS-CoV-2 infection in caspase 1/11/12/RIPK3 KO mice and compared the results with those from caspase 1/11/12/8/RIPK3 KO mice, and found that the C1/11/12/8/R3 KO did not offer better protection when compared to C1/11/12/8/R3 KO in mice in terms of survival and cytokine expression. The authors concluded caspase-8 played a key role in the induction of inflammatory response upon SARS-CoV-2 infection, which is independent of apoptosis induction.

The strength of the study is on the use of multiple KO mouse models with gene deletion.

The weakness of the study is the lack of in-depth mechanistic investigation beyond preliminary phenotypic characterization of viral replication and cytokine markers.

Comments:

1. The current study aims to study the role of caspase-8 in inflammatory response and pathogenesis in the context of SARS-CoV-2 infection. Here, all caspase-8 KO in mice was accompanied with RIPK3 KO to prevent embryonic lethality. However, RIPK3 KO itself can perturb the inflammatory pathway in multiple ways. In order to sustain the major claim of the manuscript, the authors should perform conditional caspase-8 KO to circumvent embryonic lethality.

2. All experiments were performed in mice using a N501Y-carrying mouse-adapted mutant SARS-CoV-2 virus. The major claims should be validated using human airway models such as primary airway epithelial cells or lung organoids to ensure

the findings are relevant to human physiology.

3. There is no investigation on how caspase-8 was upregulated upon infection and how caspase-8 modulated the inflammatory response.

4. Figure 1G, 4E, S1C, and S3. Gene expression in mock-infected WT, C8/R3 KO, and R3 KO mice should be shown.

5. Figure 1A and 1C. Why was the TCID50 in WT mice so different in Fig 1A and Fig 1C?

6. Figure 1G and 4E. Why did C1/11/12/8/R3 KO offer substantially less protection than C8/R3 KO?

7. Line 651. Teewn-80 should read Tween-80.

8. Figure 6E. One of the loading control lanes is missing.

Version 1:

Reviewer comments:

Reviewer #1

(Remarks to the Author)

The authors have mostly addressed my concerns. The manuscript will be ready for publication contingent to the following:

1. In the manuscript, caspase inhibition leads to increased disease and viral load. But when the non-apoptotic function of caspase 8 is inhibited, there is less cytokine expression. This is a very complex comparison as one could argue that the decreased cytokines in Emricasan-treated mice were contributing to disease severity after infection. Also, I was expecting that caspase inhibition would attenuate disease as shown by previous studies. However, the specific effects of inhibiting individual caspases could very well be different. Therefore, a discussion to compare and contrast the inhibition of caspase 8 vs. that of other caspases from existing studies should be included, if not already present. This will inform the readers the diverse and unique roles of caspases and specific effects upon inhibition. A model figure would also help to explain the authors' finding and reconcile the unexpected phenotype of caspase 8-specific inhibition.

2. Despite the efforts made by the authors to pinpoint the unique roles of caspase 8 in SARS-CoV-2 pathogenesis, the model systems used do not confidently indicate the function of caspase 8. However, due to the limitation of the model systems and the presence of control data, I think that the authors' conclusion are mostly justified. Limitations of the current study, including the use of KO strains, non-specific inhibition of caspases should be included, if not already present.

Reviewer #2

(Remarks to the Author)

Overall, I think the manuscript is improved but there are still unaddressed issues that are problematic.

Major:

1. I still think the DN lymphoproliferative T cells and overall consequent systemic inflammation observed at baseline in Ripk3/Casp8-DKO mice is still a problem in these studies. Although the authors acknowledge this (176-178), they should actually document the percent of DN B220+ T cells in lymph nodes of their mice and actually weigh and photograph lymph nodes from the corresponding DKO, Ripk3-KO, and WT mice to substantiate their claim that the mice do not have detectable lymphadenopathy (claimed in 173-176). Frankly, I would be very surprised if there were truly no detectable lymphadenopathy in these mice.

2. The comparison of infection phenotype of C1/11/12/8/R3-/- mice (Fig 5) to that of R3/C8-/- (Fig 1) is still inappropriately done and messy. These appear to be separately performed experiments and this comparison just cannot be made in this manner. If the authors want to make this comparison, then they need to perform an experiment that compares these genotypes head-to-head.

Minor:

-“Conclusion that RIPK1 does not compensate for loss of RIPK3” lines 162-163 not justified. Genetic experiments would have to be done to actually prove this statement (i.e. Ripk3-/- Casp8-/- Ripk1-KD KI).

-Fig 1c, d – missing Ripk3-/- mice for survival and weight loss

-Histology data in Fig 1h should be quantified across mice

-2g-i needs statistics, hard to tell what differences are meaningful in these bar plots; also line 301 “Fig. g-i” needs to reference fig number as well as letter

-Lines 305-318 are hard to follow because corresponding data in figures not referenced

-Heading in lines 320-321 is not justified – I agree that TNF clearly plays a more important role than IFN γ in this infection model, but to refer to this TNF-dependence as “Casp-8 driven severe disease...” is overstated since no linear genetic

dependence on TNF-Casp8 is proven in this paragraph.

-Lines 420-422 – authors say C1/11/12/R3-/- mice show significantly increased viral burden but Fig 5a says 'ns' for this comparison.

-Typo in line 685 "NF4BP1"

-Sentence starting "Moreover..." lines 696-697 doesn't seem justified simply by a comparison to influenza.

Reviewer #3

(Remarks to the Author)

1. Since the authors were not able to conduct caspase-8 KO, all results were derived from compound KO models involving casp8 KO, R3 KO, and in some case additional KOs.

The impact of casp8 was derived from comparing C8/R3 KO with R3 KO and other KO models. As the authors mentioned in reviewer#3/comment#6, "it is plausible that the additional deletion of the inflammatory caspases -1/-11/-12 in these quintuple knockout mice impairs certain compensatory mechanisms", this reviewer cannot exclude additional mechanisms being introduced with the compound KO models.

Evidence on the role of caspase 8 should be established directly instead of by eliminations.

2. Human models were requested because of the difference between human and mouse physiology. For example, RIPK1 is required for ZBP1-driven necroptosis in human cells but not mouse cells (PMID: 39982916). The authors did not provide enough evidence to show their central findings are found in human systems.

3. Mechanism of caspase-8 remains unclear. The authors suggest it is upregulated through a paracrine mechanism but non-transcriptional. This is very puzzling.

Version 2:

Reviewer comments:

Reviewer #2

(Remarks to the Author)

i support publication of the revised manuscript

Point-by-point response to comments from the reviewers of S Bader et al #NCOMMS-24-73423

Nature Communications #NCOMMS-24-73423

Original submission: Bader et al. “Non-apoptotic caspase-8 is critical for orchestrating exaggerated inflammation during severe SARS-CoV-2 infection.”

We would like to thank the Editor and the Reviewers for their insightful and constructive comments about the work presented in our manuscript. We believe we have addressed all the points raised by the reviewers in our revised manuscript that includes substantial new supporting data.

We have highlighted changes in the manuscript in yellow, and we provide the following point-by-point response to address the comments from the reviewers.

Below we paraphrase reviewer and editorial comments, *in italics and bold*, and our responses are in plain script.

REVIEWER COMMENTS

Reviewer #1 (Remarks to the Author):

Programmed cell death (PCD) and caspase activation has been shown to contribute to severe disease after coronavirus infection. Pro-inflammatory responses induced after SARS-CoV-2 infection contribute to PANoptosis in myeloid cells, necroptosis in infected epithelial cells and sequential activation of PANoptosis in uninfected bystander epithelial cells. The detrimental effects of pro-inflammatory responses after SARS-CoV-2 infection has been partially attributed to PCD mediated by caspase activation. Caspases have been reported to contribute to severe disease mostly by mediating PCD. Caspase 4/11 was reported to promote severe disease independent of PCD. Similarly, Bader et al. showed that caspase 8 contributes to heightened inflammatory response after SARS-CoV-2 infection. The authors demonstrated by genetic knockout (KO) mice, that caspase 8 promotes severe disease by cleaving negative regulator of NF- κ B N4BP1. While most of the experiments are technically sound, I have a few major concerns.

While caspase 8 KO mice are embryonic lethal, the use of C8/R3^{-/-} mice may confound the author's conclusion a RIPK3 activates necroptosis as noted by the authors, where necroptosis was reported to contribute to severe disease in infected animals (PMID: 38996010). The phenotype observed by the authors using R3^{-/-} mice does not agree with this previous study. It is possible that RIPK1 may compensate the role of RIPK3. The authors should characterize RIPK1-mediated necroptosis in R3^{-/-} mice to rule out that the phenotype observed is due to reduced necroptosis in R3^{-/-} mice.

We thank the reviewer for highlighting this study, which offers valuable insights into cell death regulation using single-cell technologies. However, we would like to clarify that the cited work was conducted exclusively in vitro, and while informative, such in vitro systems do not fully recapitulate the complex immune and tissue context of an in vivo infection. It is well established that cultured cells are more susceptible to cell death than the same cells within the whole organism, and we have previously shown that findings from in vitro necroptosis assays do not necessarily translate to the physiological setting. In our prior work (Bader et al., Cell Death & Disease, 2024), we comprehensively investigated the role of necroptosis in SARS-CoV-2 infection induced pathogenesis. Using *Mkl1*-deficient mice, we found that necroptosis is dispensable for both disease severity and viral replication in vivo (see screenshot of relevant data from that paper on the right-hand side). Additionally, we demonstrated that the elevated cytokine production characteristic of severe disease occurs

independently of necroptotic signalling (Bader et al., Cell Death & Disease, 2024). These findings support the notion that alternative, necroptosis-independent pathways mediate inflammation and tissue damage in this model of SARS-CoV-2 infection induced disease.

We appreciate the reviewer's suggestion regarding potential compensation by RIPK1 in RIPK3-deficient mice. To directly address this, we examined RIPK1 phosphorylation in lung tissue from *Ripk3*^{-/-} mice (*R3*^{-/-}) and we did not detect evidence of RIPK1 activation (see below and Fig. S1a). This is consistent with our broader dataset showing that canonical necroptotic signalling is not engaged in our SARS-CoV-2 infection model. Importantly, our observation that *R3*^{-/-} mice do not exhibit the protective phenotype seen in *C8/R3*^{-/-} animals further argues against a role for RIPK3-mediated necroptosis (as well as any other RIPK3 induced processes) in driving severe disease in this model.

While we acknowledge that all model systems have limitations, we believe our in vivo data provide a critical advance on published in vitro studies, and collectively help refine our understanding of host-pathogen interactions in SARS-CoV-2 infection.

2. The authors partially addressed point 1 by using caspase inhibitor. However, a broad spectrum caspase inhibitor was used. The phenotypic observed maybe contributed by the inhibition of other caspases. For example, inhibition of caspase 6 has been shown to protect against SARS-CoV-2 infection (PMID: 35922005). To rule out the contribution of other caspases, the authors should use caspase 8 specific inhibitor such as Z-IETD-FMK.

We thank the reviewer for this thoughtful suggestion. As requested, we performed additional experiments using the relatively caspase-8 selective inhibitor Z-IETD-FMK, following a previously established in vivo dosing regimen (Silva et al, *J Immunol* 2005). However, it is important to note that Z-IETD-FMK, while often described as a caspase-8-specific inhibitor, is not entirely selective for caspase-8. In particular, caspase-6 has been shown to efficiently cleave the IETD motif (McStay et al., *Cell Death Differ.*, 2007, Poreba et al, *Cold Spring Harb Perspect Biol.*, 2013). Moreover, the pharmacological properties of Z-IETD-FMK are not as good as those of Emricasan. These considerations limit the utility of Z-IETD-FMK for dissecting specific roles of caspase-8 in vivo. In contrast, our genetic models employing gene-targeted deletion of caspase-8 provide far greater specificity and clearly demonstrate that loss of caspase-8 improves outcomes during SARS-CoV-2 infection. However, consistent with our earlier data using the broad-spectrum caspase inhibitor Emricasan (which we now include in Fig. S1e-f), treatment of SARS-CoV-2-infected mice with Z-IETD-FMK led to increased viral burden and exacerbated weight loss, likely due to aberrant necroptosis activation (Fig. S1h-i and below). However, unlike Emricasan, Z-IETD-FMK did not suppress inflammatory cytokine production in infected mice (Fig. 1j-k and below). This difference between the effects of Emricasan vs. the effects of Z-IETD-FMK likely reflects differences in target engagement and pharmacological properties. Notably, Emricasan has been shown to effectively inhibit caspase-8 activity in the context of the caspase-8/cFLIP_L heterodimer (Brumatti et al., *Sci Transl Med.*, 2016), a key complex involved in non-apoptotic, pro-inflammatory signalling. Whether Z-IETD-FMK can also exert such an effect is not known. Taken together with our genetic data, these pharmacological findings support our central hypothesis: that inflammation during SARS-CoV-2 infection is driven by aberrant caspase-8 activity, specifically through the p41/p43 processed form of caspase-8 in complex with cFLIP_L. The new experiment using Z-IETD-FMK was therefore informative in further delineating how caspase-8 drives inflammation during infection with SARS-CoV-2, reinforcing the importance of the caspase-8/cFLIP_L complex in driving pathogenic inflammation.

3. The section heading in line 227 is logically flawed. You cannot say that C8 is not driving apoptosis (CC3 still present in Fig. 3D) and at the same time saying C8-driven apoptosis does not significantly contribute to disease. It should be changed to "apoptotic-independent C8 functions contribute to...." or something similar.

We thank the reviewer for this comment. The section heading in question, "Caspase-8 driven apoptosis does not significantly contribute to SARS-CoV-2-induced disease," was intended to convey that the apoptotic function of caspase-8 is not a major driver of disease pathology in our model. However, we agree that the original phrasing may imply the absence of apoptosis altogether, which is not accurate. As noted by reviewer #1, our data show the presence of some cleaved caspase-3 in both WT and *C8/R3^{-/-}* mice after infection with SARS-CoV-2. This shows that this relatively low level of apoptosis occurs irrespective of the presence or absence of caspase-8; i.e. caspase-8 deficiency does not markedly alter apoptotic activity in the lungs of mice after infection with SARS-CoV-2. To avoid any potential misinterpretation, we have revised the section heading to: "Apoptosis-independent functions of caspase-8 contribute to SARS-CoV-2-induced disease." This more precisely reflects our findings and the interpretation that caspase-8 contributes to SARS-CoV-2 infection induced disease through non-apoptotic mechanisms.

4. I do not agree with the author's interpretation that the level of CC3 is the same in Fig. 3D between WT and *C8/R3^{-/-}* mice. I am not sure this difference would translate to any phenotypic difference but there is a clear visual difference in the band intensity. Uninfected *C8/R3^{-/-}* mice should also be shown. Also, the authors should show that the apoptosis observed in *C8/R3^{-/-}* mice are mediated by intrinsic apoptosis pathway to support their claim.

To address the concern regarding visual differences in cleaved caspase-3 (CC3) levels and to clarify our interpretation of these results, we have repeated the western blots and included extracts from lungs of uninfected *C8/R3^{-/-}* mice, as well as *Tnf^{-/-}* mice. We have previously implicated TNF in SARS-CoV-2 infection induced pathogenesis in our model and comparing signalling in these animals together with the examination of infected *C8/R3^{-/-}* mice has now shown that the protection from severe disease afforded by the absence of caspase-8 is not fully recapitulated by the absence of TNF. These new blots are now also accompanied by densitometric quantification to allow for more objective comparisons of CC3 levels. Interestingly, closer examination of caspase-3 processing revealed incomplete processing of caspase-3: while the initial p19 fragment was present, the p17 cleavage product was largely absent in infected samples. This suggests that full execution of caspase-3-dependent apoptosis may be impaired in mice lacking caspase-8. Instead, we observed consistent caspase-7 activation and PARP cleavage, indicating that caspase-7 may be more important than caspase-3 in the execution of apoptosis in this context. Importantly, this distinction between the different functions of caspase-8 does not alter our central conclusion. Both *C8/R3^{-/-}* and *Tnf^{-/-}* mice, which exhibit similar levels of apoptosis in their lungs after infection, are protected from severe SARS-CoV-2 infection induced disease. This implies that the relatively small amount of apoptosis observed in the lungs, whether mediated by caspase-7 or incomplete caspase-3 processing, is not a major determinant of disease severity. Quantitative analyses confirmed that cleaved caspase-3 (p19) and

cleaved caspase-7 levels in lungs are comparable between SARS-CoV-2 infected WT and *C8/R3*^{-/-} mice. Please refer to the panels below and updated Figs. 4 d-f. The text has been updated accordingly.

To further investigate apoptosis in this context, as suggested by the reviewer, we performed additional analyses focusing on key components of the intrinsic apoptotic pathway. These experiments revealed no significant differences in the levels or processing of caspase-9 or BID cleavage in the lungs between SARS-CoV-2 infected vs mock mice of any of the genotypes examined. This indicates that intrinsic apoptosis is not a major driver in this model. MCL-1 expression remained at similar levels across all groups; however, *C8/R3*^{-/-} mice displayed elevated levels of pro-apoptotic BIM and anti-apoptotic BCL-XL in the lungs, irrespective of SARS-CoV-2 infection (see panels below and Fig. S3a-b). Collectively, these findings suggest that while a small level of apoptosis is evident in the lungs of SARS-CoV-2 infected mice, it proceeds through caspase-8-independent mechanisms, culminating in caspase-7 activation and PARP cleavage, and is not associated with disease severity.

5. Why did the authors choose to use *C1/11/12/8/R3*^{-/-} mice instead of *C8/R3*^{-/-} mice for the experiments shown in Fig. 4I-J and Fig. 5?

We thank the reviewer for this important question. Our rationale for using *C1/11/12/8/R3*^{-/-} and *C1/11/12/R3*^{-/-} mice was to specifically dissect the contribution of caspase-8 independently of the inflammatory caspases-

1/-11/-12. Given the extensive interconnectivity of the different programmed cell death pathways, particularly amongst initiator caspases, we aimed to minimize potential compensatory or confounding effects by using these more complex knockout strains. By comparing *C1/11/12/8/R3^{-/-}* to *C1/11/12/R3^{-/-}* animals, we ensured that the observed phenotypic and transcriptomic differences were attributable specifically to the presence or absence of caspase-8, excluding any possibility of compensatory or otherwise interfering mechanisms due to caspase-1/-11/-12 activation. This strategy allowed us to clearly identify processes and effects that are caspase-8-dependent, but independent of caspases-1/-11/-12. We have expanded the current analysis to show the upregulated and downregulated cellular pathways in each comparison more clearly (Fig. 5g-h). In this reanalysis it becomes clear, for example, that tissues in infected *C1/11/12/8/R3^{-/-}* mice have downregulated IL-1 signalling compared to the tissues from infected *C1/11/12/R3^{-/-}* mice. This further supports our hypothesis that caspase-8 is the main driver of IL-1 release during SARS-CoV-2 infection. These data are also complemented by our previously published results showing infection outcomes in *C1/11/12^{-/-}* mice (Bader et al, *Cell Death and Differentiation*, 2025 and answer to reviewer 2 below).

To further substantiate our conclusions and address the reviewer's concerns, we have now expanded our data to include *C8/R3^{-/-}* mice in this analysis. Specifically, we performed histological comparisons with WT and *R3^{-/-}* mice, conducted bulk proteomic profiling as well as spatial transcriptomics across the three relevant genotypes. These new datasets support the findings derived from our transcriptomic analyses and confirm that the phenotypic outcomes are consistent in the *C8/R3^{-/-}* background, reinforcing our original conclusions that the critical role of caspase-8 in driving inflammation during SARS-CoV-2 infection is independent of the inflammatory caspases -1/-11/-12.

6. To show that the detrimental effects of caspase 8 is mediated through N4BP1 cleavage, the authors should show that *C8/R3/N4BP1^{-/-}* mice are no longer protected.

We thank the reviewer for this suggestion. We fully agree that assessing infection outcomes in *C8/R3/N4BP1^{-/-}* mice would provide strong mechanistic evidence supporting the role of N4BP1 downstream of caspase-8. Unfortunately, this specific triple knockout strain is not available in Australia, and despite outreach to international collaborators, we were unable to secure access to such mice abroad. We believe that the generation and full in vivo characterisation of this *C8/R3/N4BP1^{-/-}* complex triple KO model falls beyond the logistical and temporal capabilities and scope of the current study. Nevertheless, our data showing that caspase-8 directly cleaves N4BP1 in lungs from SARS-CoV-2 infected mice, combined with transcriptomic and proteomic evidence of dysregulated N4BP1-regulated pathways, provide compelling evidence for a functional link. We agree that future studies using *C8/R3/N4BP1^{-/-}* mice would be a valuable next step to validate the contribution of this pathway to disease.

Reviewer #2 (Remarks to the Author):

Bader et al use a SARS-CoV2 mouse model of infection to examine the non-apoptotic role of caspase-8 in the pathological inflammatory response. The topic is impactful and interesting. There are some conclusions in the text that need to be tempered given the experimental evidence used to support them. Most importantly though, some of the genetic comparisons of infected mice throughout the manuscript have major problems that weaken the authors central claims.

Major points:

- In Figure 4, the authors compare WT mice to *C1/11/12/8/R3^{-/-}* mice. This comparison is simply uninterpretable. I understand that BSL3 conditions limit the number of mice and cohorts that can be studied in 1 infection experiment, but that doesn't change the fact that comparing a quintuply KO'ed mouse to a WT is simply not appropriate. The individual KO's of these mice may have competing effects, or a subset of the KO's may have synergistic or competing effects - the possibilities that convolute interpretation are immense. Moreover, I don't feel this experiment adds more to the conclusion the

authors are already attempting to make earlier – that Casp8 has a major effect on the SARS-CoV2 inflammatory response.

We appreciate the reviewer's concern regarding the ability to interpret comparisons between WT and *C1/11/12/8/R3^{-/-}* mice. We agree that the presence of multiple genetic deletions in this model can complicate interpretation, however, we have taken several measures to ensure the robustness of our conclusions and we have revised the manuscript accordingly. Firstly, we would like to highlight that we have previously investigated the role of inflammatory caspases-1, -11, and -12 during SARS-CoV-2 infection using the same P21 model (Bader et al., *Cell Death & Differentiation*, 2025). In that study, we demonstrated that IL-1 β -mediated inflammation and disease severity are not dependent on canonical inflammasome activation or the functions of caspases-1/-11/-12. This indicated that these inflammatory caspases are not essential for driving IL-1 β release in this disease. This prior work effectively demonstrates that there is no essential contribution of inflammatory caspases to SARS-CoV-2 infection induced disease and serves as a foundation for interpreting the current results (see panels below).

To directly address the reviewer's concern and clarify our experimental design, we have made the following revisions to the manuscript:

1. We now explicitly describe and cite our previous publication analysing the role of inflammatory caspases-1/-11 and -12 in SARS-CoV-2 infection induced disease to provide a clearer rationale for using a *C1/11/12*-deficient background in some of the experiments in our present study.
2. We have expanded the dataset by incorporating additional mice and pooling results from experiments from both *C1/11/12/8/R3^{-/-}* and *C1/11/12/R3^{-/-}* cohorts. This allowed us to more clearly delineate the specific effects attributable to caspase-8 loss during infection with SARS-CoV-2, while controlling for potential confounding effects caused by the absence of the inflammatory caspases -1/-11/-12. Maintaining a *C1/11/12/R3^{-/-}* background serves to reduce the influence of inflammasome caspase driven (compensatory) responses and focuses the interpretation of the findings on the role of caspase-8.

3. We agree with the reviewer that the WT vs. *C1/11/12/8/R3^{-/-}* comparison only confirms our hypothesis and does not add new mechanistic insight beyond what is already shown in the core comparisons. Therefore, we have revised and reduced Figure 5 and the corresponding text to keep the focus on the novel and mechanistically informative comparisons between WT, *R3^{-/-}*, and *C8/R3^{-/-}* mice.
4. To support the key findings from these comparisons, we have added new histological and bulk lung proteomic datasets from WT, *R3^{-/-}*, and *C8/R3^{-/-}* mice. These data strengthen our main conclusion that caspase-8 regulates SARS-CoV-2 induced inflammation in the absence of RIPK3, while our data from the analysis of *C1/11/12/8/R3^{-/-}* and *C1/11/12/R3^{-/-}* mice excludes confounding influences of the pro-inflammatory caspases -1/-11 and -12.

We still believe that the inclusion of data from *C1/11/12/8/R3^{-/-}* mice are valuable, particularly in light of the extensive crosstalk among programmed cell death pathways including amongst the different initiator caspases (Doerflinger et al., *Immunity* 2020). The combined deletion of caspases-1, -11, and -12 in our model is therefore designed to remove potentially confounding effects from such compensatory mechanisms, allowing us to more specifically investigate caspase-8-dependent effects. This rationale is now more clearly described in our revised manuscript and is also supported by our response to Reviewer #1 (Comment 5). We hope that the restructuring of the manuscript and clarification of our experimental strategy adequately address the concerns from Reviewer #2 while retaining the value of these complementary genetic models in dissecting caspase-8-dependent mechanisms in SARS-CoV-2 induced pathogenesis.

- Statement in lines 167-169 should be tempered. It's highly unlikely that CD3+B220+ DN T cells haven't developed at all in Ripk3/Casp8-DKO mice beyond 11 weeks. It might be that these cells haven't accumulated yet to the point that gross lymphadenopathy is readily observable, but the cells are undoubtedly there. As such, the authors should modify this statement to take into account that there could still be effects from these cells on the mice.

We thank the reviewer for raising this point. We agree that while gross lymphadenopathy and overt immune dysregulation are typically not observed in *C8/R3^{-/-}* mice on a C57BL/6 background until 12–16 weeks of age, it is possible that small populations of aberrant TCR $\alpha\beta$ ⁺CD3⁺B220⁺CD4⁻CD8⁻ double-negative T cells may be present earlier, even in the absence of macroscopic lymphoid pathology. While we cannot exclude a minor impact of these cells to immune homeostasis, our experiments were specifically designed to avoid the confounding effects of established lymphoproliferative disease, and *C8/R3^{-/-}* mice were used before any observable signs of lymphadenopathy and other clinical manifestations of LPR associated disease. We have revised the manuscript to more cautiously reflect this limitation:

“A hallmark of LPR is immune dysregulation, including the accumulation of atypical TCR $\alpha\beta$ ⁺CD3⁺B220⁺CD4⁻CD8⁻ double-negative T cells in peripheral lymphoid tissues. Although lymphoproliferative disease can complicate interpretation in studies using Caspase-8/RIPK3 compound knockout mice, we exclusively used animals prior to the development of gross lymphadenopathy. Nonetheless, we acknowledge that early populations of DN T cells may still be present and could subtly influence immune responses.”

- In Fig 1A/B vs 1C/D – why were WT, Ripk3-KO, and Ripk3/Casp8-DKO not infected in parallel? This is a bit concerning because the WT mice have TCID50 values differing by 10-100x in 1A vs 1C and similarly the average weight loss for WT in 1B is ~7% whereas it's >10% for WT in 1D. This makes it difficult to exclude the lack of an effect for Ripk3-KO alone as compared to a significant effect for Ripk3/Casp8-DKO.

We thank Reviewer #2 for highlighting this important topic. The TCID50 and weight loss data presented in Fig. 1 were derived from multiple independent experiments, each including the adequate internal controls. We usually perform at least 3 independent experiments, but not all of them include $R3^{-/-}$, and $C8/R3^{-/-}$ mice together. In our SARS-CoV-2 infection mouse model, age-matching is critical, as mice born more than 4 days apart can show measurable differences in disease severity and viral loads. Due to the logistical constraints of breeding and aging large numbers of mice from several gene knockout lines in parallel, we occasionally separated experimental groups across different time points to ensure appropriate biological replicates and to prioritize age-matching of mice within each comparison. Once experiments are pooled, it is true that viral load and disease severity can vary across replicates, as infectivity and host response are influenced by cohort-specific variation.

Nonetheless, to directly address this concern from Reviewer #2, we performed an additional age- and sex-matched infection experiment in which WT, $R3^{-/-}$, and $C8/R3^{-/-}$ mice were infected in parallel (see below, left side). Although we observed a consistent trend towards reduced viral load and weight loss in $C8/R3^{-/-}$ compared to WT and $R3^{-/-}$ mice, the reduction in TCID50 values did not reach statistical significance in this smaller cohort due to limited numbers of aged-matched animals. However, when pooled with the existing data, the combined analysis confirms the significant impact of caspase-8 deletion on viral loads (see below, right side). We have now updated the figure panels to reflect this pooled dataset and clarified this in the methods section:

“To ensure robustness, viral load and weight loss data were pooled from multiple independent infections, each including internal controls. Given the age-dependence of disease severity in this model, experimental cohorts were prioritized for age-matching even when genotypes could not be compared in the same run.”

- In addition, main Figures 1A-D and G: $C8/R3^{-/-}$ vs WT (1A-B) and $R3^{-/-}$ vs WT (1C-D) are distinct experiments, yet ELISA data in 1G appears to be a single experiment. If an experiment was performed with all 3 genotypes, thus using the same WT mice, for 1G, this data would be more effective in place of 1A-D. There is an analogous issue in Main Fig. 4A-D and E.

Please see detailed response above. In our revised manuscript we have included updated panels from $C1/11/12/8/R3^{-/-}$ and $C1/11/12/R3^{-/-}$ mice as well.

- Statistical significance in 1G?

Statistical tests were originally included in the supplemental information, but we have now additionally included them in the figure panels.

- In 1K, emricasan treatment of mice – why don't the mice fare worse with emricasan treatment since there's clearly bountiful TNF and the addition of emricasan would be predicted to trigger necroptosis? Also, the authors do not prove that emricasan's effect is via casp8, thus line 203 conclusion should not specifically make a conclusion about caspase-8 versus other caspases.

We thank Reviewer #2 for this thoughtful comment and fully agree with the interpretation. In the original version of the manuscript, we did not include data on weight loss or viral titers from the Emricasan-treated cohort. We have now added these data and confirm that WT animals treated with Emricasan exhibited increased weight loss compared to vehicle-treated controls, consistent with enhanced necroptosis due to caspase-8 inhibition and the presence of elevated TNF (see right hand side and updated Fig. S1e-f). These findings support the notion that caspase inhibition sensitizes to necroptotic cell death, particularly in the inflammatory context of SARS-CoV-2 infection. We have also updated the text accordingly.

- Lines 205-206 make a concluding sentence that Casp8 drives severe disease dependent on TNF, but the underlying data only show that Casp8 or TNF loss separately affect COVID19 disease and do not necessarily support a single linear pathway by which they influence disease. As such, this conclusion should be modified.

We understand this concern from Reviewer #2 regarding the interpretation of caspase-8 and TNF mediated effects as a single linear pathway. In response, we have now included additional data to better define the relationship between TNF and caspase-8 activation during SARS-CoV-2 infection. Specifically, we have incorporated samples from *Tnf*^{-/-} mice into our western blot analyses (Figs. 4 and 7). This revealed that the apoptotic response in the lungs from these mice after infection with SARS-CoV-2 closely resembles that observed in the lungs from infected *C8/R3*^{-/-} mice. Importantly, we find that the increase in caspase-8 in the lungs observed in WT mice after infection with SARS-CoV-2 is reduced in mice deficient for TNF. This indicates that TNF contributes to, but is not solely responsible for, the increase in caspase-8 (and presumably its activation) during infection.

These findings support a model in which in infected lungs TNF is a critical, albeit not the only, upstream activator of caspase-8 driven inflammation mediated through cleavage of N4BP1, a suppressor of NF-κB signalling. However, we acknowledge that other TNF family ligands and their cognate death receptors (e.g FAS ligand and FAS) may also contribute to this process, and we have now revised the manuscript text accordingly to reflect this more nuanced interpretation.

- Fig 4 is not clearly interpretable given the lack of single KO controls

We appreciate the reviewer's concern and would like to reiterate that we have addressed this point both in our response to Reviewer #1 (Point 5) and in an earlier response to Reviewer #2 (see above). To clarify further, we have previously performed SARS-CoV-2 infections in single KO animals relevant to this study. Specifically, in Bader et al., *Cell Death & Differentiation* (2025), we compared WT with Caspase-1/11/12 triple knockout mice and found no significant differences in disease outcomes following SARS-CoV-2 infection. Furthermore, Figure 1 of the current manuscript includes data from $R3^{-/-}$ single gene knockout mice. Together, the findings from these studies provide the necessary knockout controls for interpreting the data from the $C1/11/12/8/R3$ compound mutant mice. We have updated the text accordingly to further clarify this.

- P50 deletion does not 'abrogate' nfkb signaling and may lead to complex effects including gain-of-function dimerization events among other canonical nfkb TFs like P65 or cRel. These experiments are thus not conclusive as to nfkb signaling.

We thank Reviewer #2 for this important clarification. We agree that NF- κ B signalling is a complex and highly regulated network, and that deletion of $p50$ ($Nfkb1$) does not silence the entire NF- κ B pathway. Our intention was not to imply complete loss of NF- κ B signalling by the genetic removal of $p50$, but to assess the functional consequences of disrupting one major NF- κ B subunit within the overall NF- κ B regulatory network. To address the reviewer's point, we have revised the manuscript to temper our interpretation and avoid overstatement. We now refer to these experiments as reflective of "perturbed NF- κ B signalling caused by the removal of $p50$," rather than "abrogated signalling":

"To assess whether NF- κ B signalling is essential in driving disease in our model, we infected NF- κ B1 knockout ($p50^{-/-}$) mice with SARS-CoV-2. Compared to WT controls, $p50^{-/-}$ animals exhibited significantly reduced viral loads and attenuated weight loss at 3 dpi (Fig. 7a–b). While $p50$ deletion disrupts canonical NF- κ B signalling, it does not abolish pathway activity entirely. Therefore, our results should be interpreted as reflecting a perturbation, rather than complete loss, of NF- κ B signalling, underscoring the importance of intact NF- κ B1 function in mediating SARS-CoV-2–induced pathology in this model."

- Figures 5A and B: Labeling on top of graphs is confusing. 5A is labeled WT vs C1/11/12/8/R3-/- with WT functioning as the numerator in the fold change calculation. However, 5B is labeled C1/11/12/8/R3-/- vs. C1/11/12/R3-/- with C1/11/12/R3-/- functioning as the numerator in the fold change calculation. This should be corrected to ensure clearness. Furthermore, given that the focus is on differences due to CASP8 presence, I would recommend that analyses compare C1/11/12/8/R3-/- vs WT or C1/11/12/R3-/- (thus C1/11/12/8/R3-/- would be the numerator for fold change calculations) to demonstrate how CASP8 loss upregulates or downregulates responses to SARS-CoV-2 infection.

We thank Reviewer #2 for this helpful comment. Indeed, the label in Fig. 5a was inadvertently reversed: the comparison shown was $C1/11/12/8/R3^{-/-}$ vs WT, in line with Fig. 5b, which compares $C1/11/12/8/R3^{-/-}$ vs $C1/11/12/R3^{-/-}$. We have now corrected the labelling in Fig. 5a to reflect this appropriately. Importantly, we agree with the reviewer's point that fold changes should be expressed relative to the $C1/11/12/8/R3^{-/-}$ genotype to highlight the effects of CASP8 deletion. Additionally, we acknowledge that conventional GO enrichment analysis does not indicate the directionality of regulation. To improve interpretation of results and consistency with the new proteomics data presented in Fig. 2, we have re-analysed the RNA-seq datasets to stratify pathways based on up- and downregulated gene sets for each comparison. This updated analysis more clearly illustrates the transcriptional changes associated with CASP8 deletion in the context of SARS-CoV-2 infection (Fig. 5i-j).

- Figure 5C: Label on top of figure should be reversed to read 'Infected vs. Mock' as the figure describes proteins upregulated and downregulated in infection compared to mock.

We appreciate that Reviewer #2 identified this labelling error. We have fully updated the proteomics analysis in Fig. 2 to incorporate data from *C8/R3^{-/-}* and *R3^{-/-}* mice, providing a more complete comparison of infection-driven responses across the different genotypes of mice.

Reviewer #3 (Remarks to the Author):

In this study, Bader SM et al investigated the role of caspase-8 in inflammation and pathogenesis upon SARS-CoV-2 infection. The authors show that animal survival was improved and the expression inflammatory mediators were attenuated in the caspase-8 and RIPK3 double KO mice when compared to the WT mice. Next, they compared SARS-CoV-2 infection in caspase 1/11/12/RIPK3 KO mice and compared the results with those from caspase 1/11/12/8/RIPK3 KO mice, and found that the C1/11/12/8/R3 KO did not offer better protection when compared to C1/11/12/R3 KO in mice in terms of survival and cytokine expression. The authors concluded caspase-8 played a key role in the induction of inflammatory response upon SARS-CoV-2 infection, which is independent of apoptosis induction. The strength of the study is on the use of multiple KO mouse models with gene deletion. The weakness of the study is the lack of in-depth mechanistic investigation beyond preliminary phenotypic characterization of viral replication and cytokine markers.

We thank the reviewer for the eloquent summary.

Comments:

1. The current study aims to study the role of caspase-8 in inflammatory response and pathogenesis in the context of SARS-CoV-2 infection. Here, all caspase-8 KO in mice was accompanied with RIPK3 KO to prevent embryonic lethality. However, RIPK3 KO itself can perturb the inflammatory pathway in multiple ways. In order to sustain the major claim of the manuscript, the authors should perform conditional caspase-8 KO to circumvent embryonic lethality.

We thank Reviewer #3 for this important point. We fully acknowledge that RIPK3 deficiency on its own can modulate inflammatory signalling, and we agree that this may introduce potential confounding variables. To address this issue, we included all relevant controls using *R3^{-/-}* single knockout mice, as well as pharmacological inhibitors of caspases, to differentiate the specific contributions of the distinct functions of caspase-8 in vivo. Due to logistical limitations in Australia, including high costs and extended timelines for importing new mouse strains, we currently lack access to cell type specific conditional caspase-8 knockout mice (e.g. for removal of caspase-8 in lung epithelial cells alone, or in hematopoietic cells alone). Nonetheless, we have explored conditional strategies using the strains available to us. Specifically, we tested both the Mx-Cre and Lck-Cre systems to remove caspase-8 from type I interferon-responsive and T cells, respectively. These conditional caspase-8 knockout mice, however, did not show notable phenotypes upon SARS-CoV-2 infection (see below). We also previously discontinued the use of the LysM-Cre system for the deletion of caspase-8 or other genes of interest due to well-documented issues with leaky expression/gene deletion and limited specificity in myeloid cell populations (e.g., Stutz et al., *Immunity* 2021; Kang et al., *J. Immunol* 2004). Given these constraints and to maintain clarity of interpretation, we opted to not include these data from conditional caspase-8 deleted mice in the current manuscript.

To address cell-type contributions more precisely, we performed spatial transcriptomic profiling (Fig. 2g–k), which revealed that deletion of caspase-8, but not the deletion of *Ripk3*, selectively impaired myeloid cell recruitment to infected regions, suggesting a role of caspase-8 in shaping immune cell infiltration. This cell-type-specific effect, despite overall similar lung cell compositions across genotypes, strengthens our interpretation of a caspase-8 dependent mechanism in SARS-CoV-2 infection induced inflammation and overall pathology.

Finally, we are working with collaborators in Germany to export our P21 SARS-CoV-2 virus model to facilities with access to advanced conditional caspase-8 knockout mice and mice with point-mutations in critical residues in caspase-8. These future studies, requiring PC3 containment and animal ethics approval, will enable further dissection of the different functions of caspase-8 across different cell types during infection with SARS-CoV-2. These experiments will take 2-3 years to conduct and therefore fall beyond the scope of the present study.

2. All experiments were performed in mice using a N501Y-carrying mouse-adapted mutant SARS-CoV-2 virus. The major claims should be validated using human airway models such as primary airway epithelial cells or lung organoids to ensure the findings are relevant to human physiology.

We appreciate the suggestion from Reviewer #3 to validate our findings in human airway models such as primary airway epithelial cells or lung organoids. While we acknowledge the utility of these systems for certain mechanistic studies, our work purposefully focuses on a robust in vivo model, which we have extensively characterized and benchmarked against human COVID-19 pathology (Bader et al., *PNAS* 2023). It is important to note that many earlier studies of SARS-CoV-2 relied on simplified in vitro systems or mouse models with systemic viral dissemination and features of encephalitis that do not accurately recapitulate the predominant pulmonary pathology seen in human COVID-19 patients. In contrast, our mouse-adapted SARS-CoV-2 strain (P21) causes localized lung infection and immune-mediated disease that accurately mirrors the human condition. The strength of our in vivo mouse model lies in its ability to examine SARS-CoV-2-driven immunopathology within the complex cellular and cytokine milieu of the lung, which is inherently absent in airway epithelial cell cultures or lung organoids. Another strength of our in vivo mouse model is that we can perform the infections in a large panel of mice lacking one or several genes of interest, thereby identifying factors that are critical for pathology.

While in vitro systems offer rapid and accessible platforms, they lack key aspects of in vivo immune dynamics. In particular, interpretation of cell death pathways in culture can be misleading, as isolated epithelial cells or monocultures of immune cells exhibit heightened susceptibility to certain cell death inducing stimuli in the absence of regulatory signals from neighbouring cell types. This is also pertinent for caspase-8 dependent responses, where cytokine-mediated signalling plays a central role that is difficult to model in vitro. Indeed, in vitro studies have tended to focus on the role of caspase-8 in cell death in the response to SARS-CoV-2, while overlooking the apoptosis unrelated pro-inflammatory functions of caspase-8, which are less readily reproduced outside a whole organism. Nevertheless, to strengthen the translational relevance of our findings and to complement existing literature, we have conducted additional experiments in human cell line and human lung tissue derived studies in vitro:

1. Monocultures: SARS-CoV-2 infection of Calu-3 cells led to marked increases of the p41/p43 forms of caspase-8 at 48 hours post-infection (see below). However, we did not observe this effect in

primary human macrophages, possibly due to the absence of epithelial and stromal cell derived signals that are likely to modulate such responses in vivo.

- Human lung tissue explants in culture: To model infection in human lung tissue in culture, we infected human lung explants derived from surgical biopsies with SARS-CoV-2. These lung explants retain the native tissue architecture and immune cell composition in culture. While low infection efficiency limited the ability to quantify cytokine release, we observed an increase in caspase-8 protein in infected regions (similar to the lungs of infected mice), as confirmed by co-localization with SARS-CoV-2 nucleocapsid staining (Fig. 6e and below).
- We performed transcriptomic analysis on in vitro SARS-CoV-2 infected and mock human lung explants to compare how pathway activation in human lung tissue compared to our murine model. Analysis showed only modest transcriptional up-regulation of caspase-8 (Fig. 6f). This observation aligns with our previously published lung transcriptomic data from SARS-CoV-2 P21-infected mice. In our mouse model, caspase-8 mRNA levels remained unchanged after infection with SARS-CoV-2 compared to mock-infected mice, suggesting that caspase-8 may be increased through post-transcriptional processes. Importantly, this shows again how our mouse model of SARS-CoV-2 infection strongly correlates to data generated with experiments using human tissue in culture. In our human lung transcriptomic analysis, we observed increased expression of multiple genes involved in caspase-8 signalling, including (*cflar* encodes cFLIP), *Fas*, *Il-1 β* , *Tnf*, *Trail* and *Tnfr1* (Fig. 6f).

Furthermore, our findings are supported by several independent studies that have reported increased levels of caspase-8 in lungs of COVID-19 patients compared to lungs from uninfected healthy humans. Specifically, increased caspase-8 protein levels have been associated with more severe COVID-19 outcomes in human patients (Mao et al., *Cell Reports* 2024; Haljasmägi et al., *Sci Rep* 2020; Elhadad et al., *Thromb Res* 2023). Notably, these observations, like ours, are predominantly evident at the protein level rather than at the mRNA level, underscoring the value of post-transcriptional analysis in capturing functionally relevant changes.

In summary, while we agree that human cell-based models can provide useful mechanistic insight, our in vivo mouse model offers a unique platform for studying the immunological complexity of SARS-CoV-2-induced pathology. The inclusion of data from studies using cultured human lung explants, alongside human patient literature, collectively support the relevance of our findings to human disease. We decided not to

include data from Calu-3 cells and primary macrophages in the revised manuscript as we believe this is an inferior model to using human patient derived explanted lung tissue.

3. There is no investigation on how caspase-8 was upregulated upon infection and how caspase-8 modulated the inflammatory response.

While we agree that further dissection of the upstream signals leading to the increase in caspase-8 after infection with SARS-CoV-2 and its downstream effects on inflammation would be valuable, we have made several key observations that begin to address these questions. In Figure 7, we demonstrate that caspase-8 mediates cleavage of N4BP1, a known negative regulator of NF- κ B signalling and cytokine production. This cleavage event was observed specifically during SARS-CoV-2 infection and was reduced in both *Tnf*^{-/-} and *C8*^{-/-}/*R3*^{-/-} animals, indicating that processing of N4BP1 is dependent on both TNF induced signalling and caspase-8. These data suggest that TNF acts upstream of caspase-8 to promote its activation, most likely via TNFR1. Caspase-8 in complex with cFLIP, will then amplify inflammatory signalling by inactivating N4BP1, thereby enhancing NF- κ B-driven cytokine production. In addition, we show that the increase in caspase-8 protein occurs in cells that are not directly infected with SARS-CoV-2 (Fig. 6 and S5). This supports a model in which the increase in caspase-8 is induced through indirect, paracrine mechanisms, likely downstream of inflammatory cytokines or damage-associated signals, rather than being triggered by viral replication within individual infected cells.

Our transcriptomic analyses reveal that caspase-8 is not transcriptionally upregulated during infection with SARS-CoV-2; instead, its accumulation is observed only at the protein level. This points to a post-transcriptional or post-translational mode of regulation. Such regulation may involve cytokine-mediated signalling pathways (e.g., TNF/TNFR1 and/or other death receptors and their ligands), enhanced protein stability, or alterations in ubiquitination or proteasomal degradation. For instance, caspase-8 protein accumulation has been shown to occur in certain types of cancers (Ruiz-Ruiz et al, *J Biol Chem*, 2004, Xia et al, *Cell Death & Disease*, 2021), and while the mechanisms remain not fully understood, similar underlying mechanisms could be in place during viral infections.

Although we focused primarily on demonstrating the functional relevance of caspase-8 in driving SARS-CoV-2 infection induced inflammation and cytokine production in vivo, we agree that further mechanistic studies, e.g., dissecting the specific death receptor pathways or adaptor proteins (such as FADD or TRADD) involved in caspase-8 activation, would be of interest. Such investigations, using a panel of cell type specific knockout mice of these genes will require 2 to 3 years of work and are therefore beyond the scope of the current manuscript but represent exciting future directions.

In summary, our data suggest that caspase-8 is post-transcriptionally up-regulated during SARS-CoV-2 infection via TNF-dependent mechanisms and that caspase-8 amplifies the inflammatory response through cleavage of N4BP1 a well-known negative regulator of NF- κ B signalling. Caspase-8 likely acts in uninfected bystander cells. These findings help clarify the non-apoptotic, immune-modulatory role of caspase-8 in the context of COVID-19 disease.

4. Figure 1G, 4E, S1C, and S3. Gene expression in mock-infected WT, C8/R3 KO, and R3 KO mice should be shown.

We thank Reviewer #3 for this comment, though we were initially uncertain about the exact nature of the request. The figures referenced (Fig. 1g, 4e, S1c, and S3) present cytokine levels measured by ELISA, not gene expression data. These analyses were specifically designed to assess SARS-CoV-2 infection induced cytokine responses across different genotypes of mice, and as such, comparisons were made between infected animals, normalized to WT controls within each experimental cohort. Including mock samples in these specific comparisons would fundamentally alter the analytical framework and shift the focus from infection-induced differences between genotypes to intra-genotype baseline variability.

If the reviewer instead requests cytokine gene expression data from tissues of mock challenged WT, *R3*^{-/-}, and *C8*^{-/-}/*R3*^{-/-} mice, we would like to clarify that while transcriptional profiling can provide useful insights,

it often does not correlate well with cytokine protein abundance. This is particularly true in the context of inflammation, where cytokine expression is tightly regulated at multiple levels, not only including transcription but also mRNA stability, translation efficiency, and post-translational processing of the proteins. Given that cytokine mediated pathology operates at the protein level, and that secretion dynamics cannot be inferred from mRNA data alone, we prioritized measurement of protein abundance through ELISA and proteomic analyses as the most biologically relevant readouts.

Nevertheless, to ensure the robustness and completeness of our dataset, we have now included tissues from mock control mice in our transcriptomic and proteomic experiments. These additional data are presented in the revised Figures 2a and 4d and provide further context for interpreting SARS-CoV-2 infection induced responses in WT, $R3^{-/-}$ and $C8^{-/-}/R3^{-/-}$ mice. We hope this clarification and the inclusion of the new data from mock mice in our proteomics and transcriptomics analyses address this concern from Reviewer #3.

5. Figure 1A and 1C. Why was the TCID50 in WT mice so different in Fig 1A and Fig 1C?

We appreciate this concern from Reviewer #3. The data presented are derived from 2-3 independent infection experiments, which is our standard approach to ensure reproducibility. Due to slight variability in SARS-CoV-2 infectivity and host response across experimental cohorts, we occasionally observe differences in TCID50 values. For SARS-CoV-2 infection in mice, outcomes such as weight loss and viral load can be influenced by small age differences (even 4-7 days) or subtle batch effects in virus stocks.

To address this concern, we have now conducted and included a new repeat experiment in which WT, $C8^{-/-}/R3^{-/-}$, and $R3^{-/-}$ mice were age-matched and infected with SARS-CoV-2 side-by-side. These new data have been incorporated into the revised version of Figure 1, and we have also pooled the viral load data from all experiments to strengthen statistical power (please also see answer to Reviewer #2 above).

6. Figure 1G and 4E. Why did $C1/11/12/8/R3$ KO offer substantially less protection than $C8/R3$ KO?

We thank Reviewer #3 for highlighting this interesting observation. Indeed, $C1/11/12/8/R3^{-/-}$ animals appear to be slightly less protected from the infection-induced cytokine storm compared to $C8^{-/-}/R3^{-/-}$ animals. However, it is important to note that this modest difference between these two genotypes of mice does not substantially impact overall infection outcomes, including viral burden, weight loss, and mouse survival. The inflammatory response after infection with SARS-CoV-2 in $C1/11/12/8/R3^{-/-}$ mice remains broadly similar to that observed in $C8^{-/-}/R3^{-/-}$ mice, with key pro-inflammatory cytokines, such as IL-1 β and TNF significantly reduced compared to infected WT mice (Fig. 1g and 4e).

When we analyse viral loads across age-matched mice of several genotypes, including WT, $R3^{-/-}$, $C1/11/12^{-/-}$, $C1/11/12/R3^{-/-}$, $C8^{-/-}/R3^{-/-}$ and $C1/11/12/8/R3^{-/-}$, $C1/11/12/8/R3^{-/-}$ mice demonstrate a comparable reduction in viral burden to $C8^{-/-}/R3^{-/-}$ mice (see right-hand side). That said, the slightly diminished protection the reviewer observed in $C1/11/12/8/R3^{-/-}$ compared to $C8/R3^{-/-}$ mice could reflect a subtle protective role for the inflammatory caspases -1/-11 and -12 in antiviral defence, which may be particularly apparent in the absence of caspase-8 and RIPK3. While caspases-1 and -11 are best known for their roles in the maturation of IL-1 β as well as IL-18 and in pyroptosis, they have also been implicated in promoting pathogen clearance through modulation of immune cell function and induction of cell-intrinsic defence pathways. Therefore, it is plausible that the additional deletion of the inflammatory caspases -1/-11/-12 in these quintuple knockout mice impairs certain compensatory mechanisms that would otherwise support protection in SARS-CoV-2 infected $C8^{-/-}/R3^{-/-}$ mice.

We have now clarified this point in the revised discussion. In summary, these data suggest that caspases-1, -11, and -12 may retain a minor, potentially protective role in antiviral immunity, possibly by facilitating the clearance of infected cells or shaping the immune response.

7. Line 651. Teewn-80 should read Tween-80.

We thank the reviewer for identifying this typographical error. The term has been corrected in the revised manuscript.

8. Figure 6E. One of the loading control lanes is missing.)

We apologise for this oversight. This issue arose due to an error in figure cropping during figure preparation. We have corrected this error in the revised version of Figure 6e and have ensured that all loading control lanes are now fully displayed.

We have made substantial changes to the text figures and supplementary figures, including:

1. Fig. 1a: additional TCID50 viral titre measurements in extracts from lungs of SARS-CoV-2 infected mice side-by-side comparison of $C8^{-/-}/R3^{-/-}$ vs $R3^{-/-}$ vs. WT mice and pooled data.
2. Fig. 1b: additional weight loss data of side-by-side comparison of SARS-CoV2 infected $C8/R3^{-/-}$ vs $R3^{-/-}$ vs WT mice, and pooled data.
3. Fig. S1a: Western blots to detect RIPK1 activation (i.e. phosphorylated RIPK3) in lungs from SARS-CoV-2 infected mice.
4. Fig. 1h: histology images of lungs from SARS-CoV-2 infected mice comparing $C8^{-/-}/R3^{-/-}$, $R3^{-/-}$ and WT mice (H&E staining, IHC staining for SARS-CoV-2 nucleocapsid and F4/80).
5. Fig. S1e: histology images of lungs from SARS-CoV-2 infected mice comparing $C8^{-/-}/R3^{-/-}$, $R3^{-/-}$ and WT mice (stained for MPO and CD3).
6. Fig. 1j: cytokine measurements of SARS-CoV2 infected WT mice treated with the caspase-8 preferential inhibitor z-IETD-FMK.
7. Fig. S1j-k: viral burdens and weight loss of SARS-CoV2 infected WT mice treated with the caspase-8 preferential inhibitor z-IETD-FMK.
8. Fig. S1g-h: viral burdens and weight loss of SARS-CoV2 infected WT mice treated with the broad-spectrum caspase inhibitor Emricasan.
9. Fig. 2a-b: New figure including comparisons of proteomics analyses from lungs of SARS-CV-2 infected vs mock $C8^{-/-}/R3^{-/-}$ vs $R3^{-/-}$ vs WT mice.
10. Fig. 2c-f: Comparisons of proteomics analysis of SARS-CoC-2 infected $C8^{-/-}/R3^{-/-}$ vs $R3$ vs WT mice.
11. Fig. S2: New figure including $C8^{-/-}/R3^{-/-}$ vs $R3^{-/-}$ vs WT proteomics analysis of mock and infected lungs.
12. Fig. 2g-k: Spatial transcriptomics comparing SARS-CoV-2 infected $C8^{-/-}/R3^{-/-}$, $R3^{-/-}$ and WT mice.
13. Fig. S3a: Western blots of intrinsic apoptosis pathway proteins in WT and $C8^{-/-}/R3^{-/-}$ mice (Caspase-9, BIM).
14. Fig. S3b: Western blots of intrinsic apoptosis pathway proteins in $C8^{-/-}/R3^{-/-}$ mice (BID, MCL-1, BCL-XL).
15. Fig. 4d: Western blots of lungs of infected WT, $C8^{-/-}/R3^{-/-}$ and $TNF^{-/-}$ mice, including lungs from mock control mice.
16. Fig. 4e: Western blots of BALs of infected WT, $C8^{-/-}/R3^{-/-}$, $TNF^{-/-}$ mice, including BALs from mock control mice.
17. Fig. 4f: Quantification of lung extract western blots.
18. Fig. 4g: Quantification of BAL Western blots.
19. Fig. 5: Re-organisation of figure, moving data to supplementary figure 4.
20. Fig. 5g-h: re-analysis of RNAseq of lungs of WT, $C1/11/12/R3^{-/-}$ and $C1/11/12/8/R3^{-/-}$ to include up and down-regulated pathways.
21. Fig. S4d-e: Comparisons of RNAseq data of WT, $C1/11/12/R3^{-/-}$ and $C1/11/12/8/R3^{-/-}$. All possible comparisons are shown.
22. Fig. S5c: Histology data including caspase-8 staining of infected lungs of young and aged mice.

23. Fig. S5c: Histology data including caspase-8 staining of infected lungs of at multiple time-points post-infection with SARS-CoV-2.
24. Fig. 6e: Histology of SARS-CoV-2 infected human patient derived lung explants in culture.
25. Fig. 6f: Spatial transcriptomics data of SARS-CoV-2 infected human patient derived lung explants in culture.
26. Fig. 7e: Western blot of *C8/R3*^{-/-} and *Tnf*^{-/-} mice.
27. Fig. 7f: Quantification of western blot.

Response to Reviewer and Editor comments

Nature Communications #NCOMMS-24-73423

Original submission: Bader et al. "Non-apoptotic caspase-8 is critical for orchestrating exaggerated inflammation during severe SARS-CoV-2 infection."

We would like to thank the editors for the opportunity to submit a second revision of our manuscript. We appreciate the careful consideration of our work and the constructive feedback provided. Below we paraphrase reviewer comments, *in italics and bold*, and our responses are in plain text.

REVIEWER COMMENTS

Reviewer #1 (Remarks to the Author):

The authors have mostly addressed my concerns. The manuscript will be ready for publication contingent to the following:

1. In the manuscript, caspase inhibition leads to increased disease and viral load. But when the non-apoptotic function of caspase 8 is inhibited, there is less cytokine expression. This is a very complex comparison as one could argue that the decreased cytokines in Emricasan-treated mice were contributing to disease severity after infection. Also, I was expecting that caspase inhibition would attenuate disease as shown by previous studies. However, the specific effects of inhibiting individual caspases could very well be different. Therefore, a discussion to compare and contrast the inhibition of caspase 8 vs. that of other caspases from existing studies should be included, if not already present. This will inform the readers the diverse and unique roles of caspases and specific effects upon inhibition. A model figure would also help to explain the authors' finding and reconcile the unexpected phenotype of caspase 8-specific inhibition.

2. Despite the efforts made by the authors to pinpoint the unique roles of caspase 8 in SARS-CoV-2 pathogenesis, the model systems used do not confidently indicate the function of caspase 8. However, due to the limitation of the model systems and the presence of control data, I think that the authors' conclusion are mostly justified. Limitations of the current study, including the use of KO strains, non-specific inhibition of caspases should be included, if not already present.

Reviewer #2 (Remarks to the Author):

Overall, I think the manuscript is improved but there are still unaddressed issues that are problematic.

Major:

1. I still think the DN lymphoproliferative T cells and overall consequent systemic inflammation observed at baseline in Ripk3/Casp8-DKO mice is still a problem in these studies. Although the authors acknowledge this (176-178), they should actually document the percent of DN B220+ T cells in lymph nodes of their mice and actually weigh and photograph lymph nodes from the corresponding DKO, Ripk3-KO, and WT mice to substantiate their claim that the mice do not have detectable lymphadenopathy (claimed in 173-176). Frankly, I would be very surprised if there were truly no detectable lymphadenopathy in these mice.

2. The comparison of infection phenotype of C1/11/12/8/R3-/- mice (Fig 5) to that of R3/C8-/- (Fig 1) is still inappropriately done and messy. These appear to be separately performed experiments and this

comparison just cannot be made in this manner. If the authors want to make this comparison, then they need to perform an experiment that compares these genotypes head-to-head.

Minor:

*-“Conclusion that RIPK1 does not compensate for loss of RIPK3” lines 162-163 not justified. Genetic experiments would have to be done to actually prove this statement (i.e. Ripk3^{-/-} Casp8^{-/-} Ripk1-KD KI).
-Fig 1c, d – missing Ripk3^{-/-} mice for survival and weight loss.
-Histology data in Fig 1h should be quantified across mice.
-2g-i needs statistics, hard to tell what differences are meaningful in these bar plots; also line 301 “Fig. g-i” needs to reference fig number as well as letter
-Lines 305-318 are hard to follow because corresponding data in figures not referenced
-Heading in lines 320-321 is not justified – I agree that TNF clearly plays a more important role than IFN γ in this infection model, but to refer to this TNF-dependence as “Casp-8 driven severe disease...” is overstated since no linear genetic dependence on TNF-Casp8 is proven in this paragraph.
-Lines 420-422 – authors say C1/11/12/R3^{-/-} mice show significantly increased viral burden but Fig 5a says ‘ns’ for this comparison.
-Typo in line 685 “NF4BP1”
-Sentence starting “Moreover...” lines 696-697 doesn’t seem justified simply by a comparison to influenza.*

Reviewer #3 (Remarks to the Author):

- 1. Since the authors were not able to conduct caspase-8 KO, all results were derived from compound KO models involving casp8 KO, R3 KO, and in some case additional KOs.
The impact of casp8 was derived from comparing C8/R3 KO with R3 KO and other KO models. As the authors mentioned in reviewer#3/comment#6, “it is plausible that the additional deletion of the inflammatory caspases -1/-11/-12 in these quintuple knockout mice impairs certain compensatory mechanisms’, this reviewer cannot exclude additional mechanisms being introduced with the compound KO models.
Evidence on the role of caspase 8 should be established directly instead of by eliminations.*
- 2. Human models were requested because of the difference between human and mouse physiology. For example, RIPK1 is required for ZBP1-driven necroptosis in human cells but not mouse cells (PMID: 39982916). The authors did not provide enough evidence to show their central findings are found in human systems.*
- 3. Mechanism of caspase-8 remains unclear. The authors suggest it is upregulated through a paracrine mechanism but non-transcriptional. This is very puzzling.*

COMBINED ANSWER TO COMMENTS FROM THE EDITOR AND REVIEWERS

Reviewer #1: We thank the reviewer for recognizing the overall strength of our findings despite the inherent limitations of the available models. As requested, we have further elaborated on the complexities of caspase inhibition in viral pathogenesis and provided a direct comparison with prior studies using caspase inhibitors. The updated discussion (see below highlighted in yellow, point #3) now more clearly outlines how distinct modes of caspase-8 inhibition can lead to differential outcomes in cytokine production and disease severity.

Reviewer #2: We appreciate the detailed feedback from Reviewer #2 and have now addressed all major concerns with new experimental data, including quantification of DN B220⁺ T cells and a direct head-to-head comparison of mice of the requested genotypes. We have included additional information on the

lymphadenopathy phenotype caused by the absence of caspase-8 (in context of the additional absence of RIPK3) in the manuscript (see below highlighted in yellow, point #1). We believe these new data strengthen the manuscript considerably. Additionally, while some minor points raised exceed the current scope of this study, we have clarified figure references and adjusted language where appropriate.

Reviewer #3: We acknowledge the concerns of Reviewer #3 regarding the limitations of using compound knockout mouse models and the absence of single caspase-8 knockouts due to embryonic lethality. While genetic elimination remains the only viable and robust in vivo strategy to study the function of caspase-8, we have made efforts to substantiate and solidify our conclusions by incorporating both genetic and pharmacological approaches, each with their own strengths and caveats. Although we agree that species-specific differences in RIPK1/ZBP1 regulation do exist, we note that these pathways were not the primary focus of this study.

Guided by the editorial recommendations, we performed additional experiments and revised the manuscript to specifically address the three major points raised by the reviewers:

1. Systemic inflammation

To evaluate the presence of DN B220⁺ T cells in our mouse models, we assessed spleen and lymph node sizes in WT and knockout animals across multiple genotypes and age groups and performed flow cytometric analysis on animals from our breeding colonies. At 6 weeks of age, *C8/R3^{-/-}* and *C1/11/12/8/R3^{-/-}* mice exhibited spleen and lymph node sizes comparable to those of *C1/11/12/R3^{-/-}* and WT control mice. By 9-12 weeks, *C1/11/12/8/R3^{-/-}* mice began to display mild splenomegaly and increased lymph node sizes compared to age-matched control mice (see figure below, left panel). Flow cytometric analysis revealed an increase in the frequency of B220⁺ T cells in caspase-8-deficient mice from approximately 10 weeks of age onward (right panel), which was lower than in our *C8/R3^{-/-}* positive control of animals at 30 weeks that display physical signs of pathology.

Of note, for this revision experiment we analysed the animals we had currently available across several colonies, rather than awaiting the breeding and precise age-matching of all genotypes, which given the amount of strains studied would have taken 6-12 months to achieve. Despite this limitation, we believe the newly added data robustly illustrate the age-associated emergence of lymphadenopathy in our models. Importantly, the findings from this work reveal that at the age used for the majority of the experiments in our study on which we base our main conclusions (mice aged 6-7 weeks) no apparent underlying LPD was present. While the enlargement observed in 10-week-old caspase-8 knockout animals is subtle and not visually apparent in live animals, we have revised the manuscript to acknowledge the potential contribution of DN B220⁺ T cell proliferation in mid-aged cohorts:

“A hallmark of LPR syndrome is immune dysregulation, including the accumulation of atypical TCR $\alpha\beta$ ⁺CD3⁺B220⁺CD4⁻CD8⁻ double-negative T cells in peripheral lymphoid tissues. While this phenotype was absent in our younger cohorts (5-7 weeks), early but mild lymphoproliferative disease was detectable by 10 weeks in compound caspase-8 (and RIPK3)-deficient animals. Although this could subtly influence immune responses in the infected knockout models, the mild nature of the phenotype at this age makes it unlikely to fully account for the infection-associated differences observed.”

We additionally provide a supplementary file at the end of this document containing uncropped images of spleens and lymph nodes from multiple animals per genotype, including ruler-based size references. The representative images included below summarize the key findings. We also provide the gating strategy for our flow cytometry quantification of TCR $\alpha\beta$ ⁺CD3⁺B220⁺CD4⁻CD8⁻ double-negative T cells.

2. Side-by-side comparison of knockout animals

We thank the reviewer for their continued engagement and acknowledge the concern regarding the comparison between *C1/11/12/8/R3^{-/-}* and *C8/R3^{-/-}* mice presented in the original version of the manuscript. In our previous response, we provided viral load data from age-matched cohorts across several genotypes (WT, *R3^{-/-}*, *C1/11/12^{-/-}*, *C1/11/12/R3^{-/-}*, *C8/R3^{-/-}*, and *C1/11/12/8/R3^{-/-}*). These cohorts were pooled from several independent experiments but were carefully matched for age, reflecting our long-standing approach to achieve statistical robustness and biological reproducibility in this complex genetic model. Given the strict age dependence of disease severity in this SARS-CoV-2 infection system, assembling large numbers of precisely age-matched animals within a single experiment is often impractical. Therefore, pooling across matched cohorts has consistently provided us with reliable and representative comparisons of disease phenotypes.

However, in light of the reviewer's request for a direct, within-experiment comparison, we have for this revision conducted a new, dedicated infection study involving only age-matched *C8/R3^{-/-}* and *C1/11/12/8/R3^{-/-}* mice. Alongside viral burden data, we also provide macroscopic images of spleens and lymph nodes from all analysed animals. Due to biosafety constraints, we were unable to include precise organ weights, as no calibrated balance is available within our BSL-3 facility. Nevertheless, this new dataset clearly demonstrates that *C8/R3^{-/-}* mice exhibit significantly lower viral burden compared to *C1/11/12/8/R3^{-/-}* animals. These findings are consistent with our earlier results, in which statistical significance was not reached due to the inclusion of multiple genotypes and increased intergroup variability (see figure below).

Side-by-side infection of *C8/R3^{-/-}* and *C1/11/12/8/R3^{-/-}* animals:

Previously shown data of pooled experiments:

Left-hand-side: all pooled data.

Right hand side: same data, but only *C8/R3^{-/-}* and *C1/11/12/8/R3^{-/-}* mice.

This phenotypic difference is already discussed in the current version of the manuscript (see text below). As we are only able to speculate on the mechanistic basis for this observation, we believe further discussion at this stage would be premature:

“Interestingly, the absence of caspase-1, -11, -12 together with RIPK3 seemed to worsen SARS-CoV-2-induced disease, perhaps through a mechanism that further unleashes the pro-inflammatory function of caspase-8. When caspase-8 was removed alongside with the absence of caspases-1, -11, -12, SARS-CoV-2 induced disease was yet again substantially ameliorated. Notably, we observed a slight diminished protection from SARS-CoV-2-induced pathology in *C1/11/12/8/R3^{-/-}* compared to *C8/R3^{-/-}* animals. This may reflect a minor role for these three inflammatory caspases in the response of mice to infection with SARS-CoV-2. While caspases-1 and -11 are well known for their roles in the maturation of IL-1 β as well as IL-18 and pyroptosis, they have also been implicated in promoting pathogen clearance through modulation of immune cell function and induction of cell-intrinsic defence processes. Therefore, it is plausible that the additional deletion of inflammatory caspases in this quintuple knockout impairs certain compensatory mechanisms that would otherwise support protection in *C8/R3^{-/-}* mice.”

3. Caspase inhibition

We agree that the effects of caspase inhibition in our models are complex. To clarify: in our experiments, treatment with the broad spectrum caspase inhibitor emricasan or the more selective (yet still not highly specific) caspase-8 inhibitor Z-IETD-FMK both worsened disease severity, as evidenced by increased weight loss and/or viral load. However, only emricasan reduced pro-inflammatory cytokines, such as TNF, IL-6, and IL-1 β , whereas Z-IETD-FMK did not. This difference in results likely reflects differences in

target selectivity and pharmacological properties of the compounds used: emricasan has been shown to inhibit the caspase-8/cFLIP heterodimer, a non-apoptotic protein complex that can drive inflammatory cytokine production. Z-IETD-FMK, on the other hand, primarily targets catalytically active caspase-8 homodimers and may not inhibit the heterodimeric caspase-8/cFLIP inflammatory axis effectively. The increase in weight loss likely results from increased necroptosis, which is known to occur when caspase activity is blocked in the context of elevated TNF, a hallmark of SARS-CoV-2 infection. Therefore, it is possible that the balance between caspase-8-dependent cytokine induction and caspase-8-mediated inhibition of necroptosis determines disease outcome upon treatment with caspase inhibitors.

We mention all of these aspects in the discussion section of our manuscript, but we are unable to further expand this to include comparison of the inhibition of only caspase-8 vs. the inhibition of other caspases from existing studies, as there are currently no available inhibitors that exclusively inhibit caspase-8 without affecting other caspases (or inhibit other caspases while not impacting caspase-8). Recent studies (Bourne et al. bioRxiv 2025 doi: 10.1101/2025.02.23.639785) have again highlighted how unspecific caspase inhibitors are, making genetic knockouts the only available tool to study the effect of single caspases. In future, studies using siRNAs might be able to better answer these questions, but this currently lies beyond the scope of our study. We have adjusted our discussion as follows:

“To further dissect the contribution of caspase-8 independently of RIPK3, we examined cytokine responses following treatment with the broad-spectrum caspase inhibitor emricasan and the more selective (though not uniquely specific) caspase-8 inhibitor Z-IETD-FMK. Emricasan treatment did not alter viral replication but significantly reduced pulmonary TNF, IL-6, and IL-1 β , indicating that caspase-8-dependent cytokine release occurs independently of RIPK3. In contrast, Z-IETD-FMK failed to suppress cytokine production during infection, consistent with recent reports showing variable cytokine modulation by this inhibitor depending on cell type, pathogen, and in vivo pharmacokinetics^{83,84}. Notably, emricasan has been shown to effectively inhibit caspase-8 activity within the caspase-8/cFLIP heterodimer⁵⁶, a key driver of non-apoptotic, pro-inflammatory signalling, whereas it remains unknown whether Z-IETD-FMK achieves comparable target engagement in this context. Furthermore, recent studies have demonstrated that Z-IETD-FMK can partially inhibit other effector caspases (e.g. caspase-3/-6) and PARP, and that no currently available caspase-8 inhibitor is truly specific^{85,86}. The divergent effects observed here likely reflect differences in target selectivity, cell permeability, and tissue distribution, as well as potential off-target activities⁸⁷. Together, these findings underscore the context-dependent and multifunctional roles of caspase-8 during infection, and suggest that selective inhibition of its non-apoptotic signalling might blunt cytokine production but could also predispose to increased necroptotic cell death. Development and application of truly selective caspase-8 inhibitors or genetic silencing strategies (e.g. siRNA) will be required to formally test this hypothesis.”

Overview of spleens and lymph nodes of all animals analysed for flow cytometry

Uncropped images of spleens and lymph nodes of all animals analysed for flow cytometry

Uncropped images of spleens and lymph nodes of all infected animals